# OmniZoom: A Universal Plug-and-Play Paradigm for Cross-Device Smooth Zoom Interpolation

**Xiaoan Zhu**[*]
Zhejiang University
xiaoanzhu@zju.edu.cn

**Yue Zhao**[*]
Huawei Noah's Ark Lab
zhaoyue53@huawei.com

**Tianyang Hu**
Zhejiang University
huty@zju.edu.cn

**Jiaming Guo**
Huawei Noah's Ark Lab
guojiaming5@huawei.com

**Yulan Zeng**
Sungkyunkwan University
zengyulan@skku.edu

**Renjing Pei**
Huawei Noah's Ark Lab
peirenjing@huawei.com

**Fenglong Song**
Huawei Noah's Ark Lab
songfenglong@huawei.com

**Huajun Feng**[†]
Zhejiang University
fenghj@zju.edu.cn

## Abstract

Dual-camera smartphones suffer from geometric and photometric inconsistencies during zoom transitions, primarily due to disparities in intrinsic/extrinsic parameters and divergent image processing pipelines between the two cameras. Existing interpolation methods struggle to effectively address this issue, constrained by the lack of ground-truth datasets and motion ambiguity in dynamic scenarios. To overcome these challenges, we propose OmniZoom, a universal plug-and-play paradigm for cross-device smooth zoom interpolation. Specifically, we present a novel cross-device virtual data generation method utilizing 3D Gaussian Splatting. This method tackles data scarcity by decoupling geometric features via spatial transition modeling and correcting photometric variations with dynamic color adaptation. It is further enhanced by cross-domain consistency learning for device-agnostic semantic alignment. Additionally, we introduce a plug-and-play 3D Trajectory Progress Ratio (3D-TPR) framework that surmounts 2D spatial limitations. As components of our framework, a texture-focus strategy is introduced for high-frequency detail preservation, incorporating mask penalty constraints to suppress interpolation artifacts. Our pipeline exhibits broad compatibility with diverse interpolation methods and achieves good performance across multiple public benchmarks. Real-world evaluations on various smartphone platforms also reveal significant quality improvements after finetuning on our synthetic data, which underscores the robustness and practical effectiveness of our approach for cross-device zoom applications.

## 1 Introduction

In recent years, dual-camera systems [8, 9, 13] have become increasingly prevalent in modern smartphones, offering advanced imaging capabilities. Among these functionalities, zooming[2, 58] plays a crucial role by bridging the wide-angle and main cameras, allowing users to perceive changes in subject distance and scale without physical movements. However, performing optical zoom [10, 28]

---

[*]Equal contribution.

[†]Corresponding author.

39th Conference on Neural Information Processing Systems (NeurIPS 2025).

introduces significant geometric misalignment and photometric inconsistencies between images. Geometric misalignment arises from disparities in the intrinsic and extrinsic parameters of the dual-camera system [46, 57]. In contrast, photometric inconsistencies are caused by algorithmic divergences (e.g., white balance, color correction, and gamma correction) in Image Signal Processing pipelines (ISP) [36, 26]. These differences lead to parallax, color shifts, and exposure mismatches between the paired images, severely affecting the smoothness and realism of zoom transitions.

Notably, the transition between fixed-focus dual-camera views can be naturally formulated as an intermediate frame synthesis task. Frame interpolation (FI) [17, 25, 27, 53, 60, 47, 3] is a promising technique in this context, as it aims to synthesize temporally consistent intermediate frames between given inputs. In this work, we define **zoom interpolation (ZI)** as a specialized sub-task of FI, where the objective is to synthesize an intermediate frame that is both geometrically and photometrically consistent, given two zoomed images captured from a dual-camera system. While ZI shares the core goal of FI in generating intermediate content, it also presents several unique challenges that hinder the direct application of conventional FI methods.

First, unlike FI tasks with dense temporal supervision, ZI lacks suitable ground-truth intermediate frames. This is because deploying a continuous-zoom[2, 58] capture setup across dual-camera hardware is impractical in real-world mobile scenarios, making it difficult to train or finetune models in a supervised manner. Second, ZI suffers from severe motion ambiguity. Traditional FI approaches rely on fixed timestep maps [29, 3] to model pixel displacement, assuming smooth linear motion. However, zoom transitions often introduce complex, nonlinear object motion and parallax effects. These distort assumptions in flow estimation, leading to motion blur and structural artifacts.

To address the unique challenges of ZI, we propose OmniZoom, a universal plug-and-play paradigm for cross-device smooth zoom interpolation. This paradigm comprises two key components specifically designed for this task: **(1) Generation of cross-device smooth zoom virtual dataset.** To overcome the lack of ground-truth intermediate frames in ZI, we build upon ZoomGS [50] and adopt 3D Gaussian Splatting (3DGS) [23] to synthesize high-quality data. While ZoomGS is limited to single-device scenarios, our method further resolves cross-device geometric and photometric inconsistencies. We enhance this process with spatial transition modeling and dynamic color adaptation to decouple geometric and photometric differences between devices. Furthermore, a cross-domain consistency learning scheme ensures the generated virtual dataset maintains semantic alignment and photometric fidelity across different camera platforms. **(2) 3D-TPR framework.** To tackle spatial deficiencies and velocity ambiguity during interpolation, we propose a plug-and-play 3D-TPR module that encodes 3D motion-aware correspondence across frames. In addition, we incorporate a texture-focus strategy to preserve high-frequency details and a mask uncertainty penalty to suppress ambiguous predictions. These components work jointly to improve perceptual sharpness and mitigate artifacts during zoom synthesis.

Our pipeline is specifically designed for ZI and exhibits strong plug-and-play adaptability across a variety of FI backbones and mobile devices. By integrating the 3D-TPR framework into various existing architectures, we observe remarkable fidelity improvements and superior subjective quality on multiple public benchmarks. To further validate its real-world applicability, we construct test datasets captured from various smartphones. Finetuning FI models on our ZI dataset leads to substantial quality gains across all models on the real-world benchmark. Results demonstrate the generalizability and robustness of our approach in addressing zoom-related challenges across diverse FI models and mobile devices, providing a unified solution for achieving smooth and perceptually coherent zoom transitions in dual-camera systems.

In conclusion, our key contributions are as follows:

- We propose a novel pipeline for constructing a cross-device ZI dataset, which models geometric and photometric inconsistencies across devices. The resulting dataset enables high-quality, device-agnostic supervision, filling the gap in ground-truth data for ZI task.

- We introduce a novel ZI framework, 3D-TPR, which enhances spatial modeling via disparity-aware encoding, complemented by a texture-focus strategy and a mask penalty for better perceptual quality and robustness.

- We propose OmniZoom, a universal plug-and-play paradigm for cross-device smooth zoom interpolation that seamlessly integrates with FI networks. It consistently delivers high-quality ZI results across diverse FI models and smartphone devices in real-world scenarios.

## 2 Related works

### 2.1 Frame interpolation methods

Frame Interpolation plays a critical role in high-resolution video synthesis, slow-motion rendering, editing, and AR/VR applications [30, 33, 16]. Recent advances have explored temporal mapping strategies to improve interpolation quality. For instance, Interpany[60] replaces fixed timestep maps with 2D flow-based distance-aware index. However, its reliance on 2D optical flow limits robustness under complex 3D motion and large depth variation. We propose a 3D geometric-aware encoding mechanism that captures spatial coherence more effectively. Unlike methods such as [17] that suppress artifacts utilizing deep hierarchical features, our design handles mask uncertainty without additional architectural complexity. Despite advancements of recent learning-based FI models [3, 53, 4, 27, 19, 17, 34, 25], restoring fine textures under large motion remains challenging. We tackle this via a texture-focus training strategy that reweights reconstruction loss based on gradient cues, offering a lightweight alternative to traditional perceptual losses [20] or multi-scale losses [44].

### 2.2 3D reconstruction

3D reconstruction has evolved from classical geometry-based methods such as Structure-from-Motion (SfM) [38] and Multi-View Stereo (MVS) [14], which enable scene recovery from unstructured images to deep learning approaches that enhance robustness and generalization [12, 61, 43]. Neural Radiance Fields (NeRF) [31] and its variants (e.g., Mip-NeRF 360 [6]) achieve high-fidelity view synthesis but at high computational cost. Recently, 3D Gaussian Splatting (3DGS) [23] has emerged as an efficient alternative with explicit point-based representation and real-time rendering capability. ZoomGS [50] leverages 3DGS to create virtual cameras between dual-camera pairs for interpolation supervision, but its device-specific calibration limits generalization. In this work, we extend 3DGS and ZoomGS into a universal, cross-device virtual data generation pipeline, jointly addressing geometric and photometric inconsistencies to support consistent and smooth zoom synthesis across diverse mobile platforms.

## 3 Method

In this section, we first present a pipeline based on 3DGS [23] for constructing a cross-device smooth zoom virtual dataset. This dataset enables the generation of high-quality virtual images that are not limited to a specific camera module. Building upon this foundation, we propose the 3D-TPR framework, which enhances temporal consistency, preserves fine details, and improves spatial reliability during interpolation. The overall pipeline is illustrated in Figure 1(a).

### 3.1 Generation of cross-device smooth zoom virtual dataset

ZoomGS [50] utilizes 3DGS [23] to synthesize virtual 3D camera models $\mathcal{M}_v$ between the main camera $\mathcal{M}_m$ and wide-angle camera $\mathcal{M}_w$, facilitating the generation of pseudo-supervised training images $\tilde{I}$ via:

$$\tilde{I} = \mathcal{R}(\mathcal{M}, \mathcal{K}, \mathcal{P}), \tag{1}$$

where $\mathcal{R}$ denotes the 3DGS rendering process [23], and $\mathcal{K}$, $\mathcal{P}$ represent the intrinsic and extrinsic parameters of the corresponding cameras. However, the dataset generated by ZoomGS is device-dependent and lacks generalizability across different smartphone platforms. To overcome this limitation, we extend the ZoomGS pipeline to construct cross-device ZI datasets applicable to arbitrary smartphone configurations. As shown in Figure 1(b), we introduce a spatial transition modeling module and a color adaptation strategy that decouple the geometric and photometric characteristics of individual camera modules. Furthermore, we propose a cross-domain consistency learning approach to enforce semantic alignment and visual coherence across devices and environments.

**Spatial transition modeling.** For each scene, we capture $N$ zoom image pairs comprising wide-angle images $\{I_w^n\}_{n=1}^N$ and main images $\{I_m^n\}_{n=1}^N$. These images are then processed using COLMAP [38] to calibrate their intrinsic parameters $\{\mathcal{K}_w^n\}_{n=1}^N$, $\{\mathcal{K}_m^n\}_{n=1}^N$ and extrinsic parameters $\{\mathcal{P}_w^n\}_{n=1}^N$, $\{\mathcal{P}_m^n\}_{n=1}^N$. For each pair, we generate $M$ camera poses, including the two original cameras and $M-2$ interpolated virtual views in between. To achieve smooth spatial transitions, we interpolate both intrinsic and extrinsic parameters between the two real cameras. For the intrinsic parameters $\mathcal{K}_{v_i}$, we perform linear interpolations [15]:

$$\mathcal{K}_{v_i} = (1 - \frac{i-1}{M-1}) \odot \mathcal{K}_w + \frac{i-1}{M-1} \odot \mathcal{K}_m, \tag{2}$$

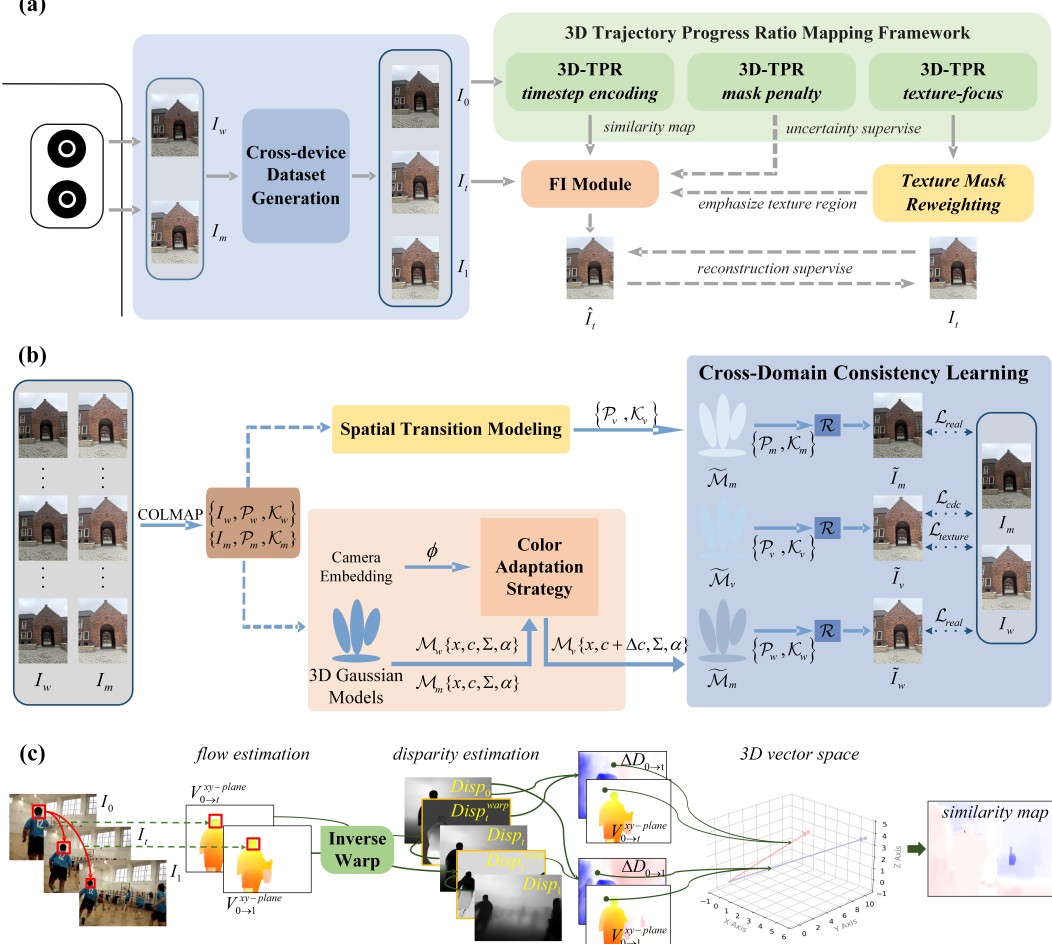

Figure 1: Overview of our proposed OmniZoom. (a) Plug-and-play integration pipeline for cross-device ZI. (b) Dual-camera ZI dataset generation via spatial-color calibration and cross-domain optimization. (c) Construction of similarity map in the 3D-TPR encoding module.

while the extrinsic parameters $\mathcal{P}_{v_i}$ are interpolated in Euclidean space [21, 5]:

$$\mathcal{P}_{v_i} = \mathcal{P}_w \odot \exp\left(\frac{i-1}{M-1} \odot \log\left(\frac{\mathcal{P}_m}{\mathcal{P}_w}\right)\right), \tag{3}$$

where $i \in 1, 2, \ldots, M-1$. This spatial transition module enables continuous and geometrically coherent virtual viewpoints by interpolating between COLMAP-estimated camera poses. It models zoom motion purely from relative transformations, without relying on any device-specific parameters.

**Color adaptation strategy.** To handle photometric inconsistencies across heterogeneous devices, we introduce a dynamic color adaptation mechanism that operates at the level of individual 3D Gaussians. Specifically, we represent each camera as a 3DGS model $\mathcal{M}$ composed of a set of 3D Gaussians. Each Gaussian is parameterized by its center position $x$, color $c$, anisotropic covariance $\Sigma$ (indicating its spatial extent), and opacity $\alpha$ [23]. To account for photometric shifts between devices, we assign each camera a learnable embedding $\phi$ (e.g., $\phi_w = 1$ for wide-angle and $\phi_m = 0$ for main camera), and interpolate the virtual camera embedding $\phi_{v_i}$ following Equation 2. A lightweight neural network, ColorNet $\mathcal{C}$, then predicts a per-Gaussian color offset based on both the geometric and appearance attributes:

$$\Delta c = \mathcal{C}(\phi, x, c, \Sigma, \alpha). \tag{4}$$

This offset is applied to each Gaussian, resulting in a view-specific virtual model $\tilde{\mathcal{M}}_{v_i}$ with updated color values $\{x, c + \Delta c, \Sigma, \alpha\}$. This learned photometric correction is continuous and device-agnostic, enabling robust color alignment across diverse camera modules. Further architectural details and ablation studies of ColorNet are provided in the Appendix A.1.

**Cross-domain consistency learning approach.** To ensure semantic and visual consistency across devices, we introduce a cross-domain consistency learning strategy. During joint optimization of the color adaptation network $\mathcal{C}$, the rendered images $\tilde{I}_w$ and $\tilde{I}_m$ are supervised by their corresponding ground truth image $I_w$ and $I_m$ via a reconstruction loss:

$$\mathcal{L}_{real} = (1 - \lambda)\mathcal{L}_1(\tilde{I}_{real}, I_{real}) + \lambda\mathcal{L}_{SSIM}(\tilde{I}_{real}, I_{real}), \tag{5}$$

where $real \in \{m, w\}$, and $\mathcal{L}_1$, $\mathcal{L}_{SSIM}$ denote the $\ell_1$ loss and SSIM loss [49], respectively. We follow the default hyperparameters from 3DGS and set $\lambda = 0.2$. However, relying solely on low-level pixel reconstruction is insufficient to ensure perceptual consistency across domains. To enhance semantic structure and texture realism, we introduce two additional feature-space constraints. We propose a semantic consistency loss $\mathcal{L}_{cdc}$ based on a pre-trained lightweight feature extractor $\mathcal{F}$ [37]:

$$\mathcal{L}_{cdc} = ||\mathcal{F}(\tilde{I}_{v_i}) - \mathcal{F}(I_{real})||_1, \tag{6}$$

Additionally, to preserve texture consistency across domains, we incorporate a texture consistency loss $\mathcal{L}_{texture}$, computed using a VGG-based texture encoder $\mathcal{T}_{texture}$ [39]:

$$\mathcal{L}_{texture} = ||\mathcal{T}_{texture}(\tilde{I}_{v_i}) - \mathcal{T}_{texture}(I_{real})||_2 \tag{7}$$

The final objective for domain-consistent virtual image generation is given by:

$$\mathcal{L}_{total} = \mathcal{L}_{real} + \mathcal{L}_{cdc} + \mathcal{L}_{texture} \tag{8}$$

### 3.2 3D-TPR framework

Traditional FI models represent the interpolation time $t$ using a fixed timestep map that is concatenated with the input frames $\{I_0, I_1\}$. However, this static encoding lacks adaptability to scene content and motion dynamics. As illustrated in Figure 1(a), our 3D-TPR framework replaces the fixed timestep map with a pixel-wise similarity map computed using 3D-TPR encoding, which captures 3D geometric relationships between the input views. In addition, we introduce a texture-focus strategy and a mask-based uncertainty penalty to provide auxiliary supervision.

**3D-TPR encoding.** As shown in Figure 1(c), $\{I_0, I_1, I_t\}$ serves as the input to the core 3D-TPR encoding module. Following InterpAny [60], we use RAFT [42] to estimate 2D pixel-wise optical flows $\{V_{0 \to t}^{xy\text{-}plane}, V_{0 \to 1}^{xy\text{-}plane}\}$. We compute the corresponding backward flows $\{V_{t \to 0}^{xy\text{-}plane}, V_{1 \to 0}^{xy\text{-}plane}\}$ using an inverse warp module $W^{-1}(\cdot)$:

$$V_{t \to 0}^{xy\text{-}plane} = W^{-1}(V_{0 \to t}^{xy\text{-}plane}), \quad V_{1 \to 0}^{xy\text{-}plane} = W^{-1}(V_{0 \to 1}^{xy\text{-}plane}). \tag{9}$$

Concurrently, we employ an efficient network Lite-Mono [54] to predict monocular disparity maps $\{Disp_0, Disp_t, Disp_1\}$ from the input frames $\{I_0, I_t, I_1\}$. The predicted disparities $\{Disp_t, Disp_1\}$ are then warped to the reference view $I_0$ using the backward flows via a simple warp operation $W(\cdot)$, yielding the warped disparities $\{Disp_t^{warp}, Disp_1^{warp}\}$:

$$Disp_t^{warp} = W(Disp_t, V_{t \to 0}^{xy\text{-}plane}), \quad Disp_1^{warp} = W(Disp_1, V_{1 \to 0}^{xy\text{-}plane}). \tag{10}$$

We then compute the disparity residuals $\{\Delta D_{0 \to t}, \Delta D_{0 \to 1}\}$ by measuring the differences between the warped disparity maps $\{Disp_t^{warp}, Disp_1^{warp}\}$ and the reference disparity map $Disp_0$:

$$\Delta D_{0 \to t} = ||Disp_t^{warp} - Disp_0||_\psi, \quad \Delta D_{0 \to 1} = ||Disp_1^{warp} - Disp_0||_\psi, \tag{11}$$

where $||\cdot||_\psi$ denotes a robust error norm. These components jointly yield the disparity variation vector $Var^z = \Delta D_{0 \to t}/\Delta D_{0 \to 1}$, which reflects the disparity differential ratio of the target frame $I_t$. We hypothesize that this progress-ratio-like scalar can serve as a learned timestep prior, replacing fixed interpolation index and enabling better temporal alignment under non-uniform motion. Consequently, we fuse the disparity residual $\Delta D_{0 \to t}$ and the planar motion vector $V_{0 \to t}^{xy\text{-}plane}$ into a unified 3D motion field $\Delta V_{0 \to t}^{xyz}$ via a cross-domain aggregation module $A(\cdot)$. This multimodal fusion is symmetrically extended to the temporal span $0 \to 1$. Each spatial location $(i, j, k) \in \mathbb{R}^3$ in 3D motion fields is defined as:

$$(\Delta V_{0 \to t}^{xyz})^{i,j,k} = A(\Delta D_{0 \to t}, V_{0 \to t}^{xy\text{-}plane}), \quad (\Delta V_{0 \to 1}^{xyz})^{i,j,k} = A(\Delta D_{0 \to 1}, V_{0 \to 1}^{xy\text{-}plane}). \tag{12}$$

The trajectory progress similarity at $(i, j, k)$ is then computed via cosine similarity:

$$Sim(i, j, k) = \frac{\Delta V_{0 \to t}^{xyz}(i, j, k) \cdot \Delta V_{0 \to 1}^{xyz}(i, j, k)}{||\Delta V_{0 \to t}^{xyz}(i, j, k)|| \cdot ||\Delta V_{0 \to 1}^{xyz}(i, j, k)||}. \tag{13}$$

This formulation effectively decouples perspective projection effects from true 3D motion, addressing a key limitation of conventional 2D optical flow methods while preserving temporal coherence across interpolated frames.

**Gradient-coupled texture-focus training strategy.** To mitigate the common tendency of FI networks to oversmooth high-frequency content [52], we introduce a texture-focus reweighting scheme. We identify texture regions in the ground-truth frame $I_t$ using a multi-scale, kernel-based detector $\mathcal{T}$:

$$M = \mathcal{T}(I_t) \in \{0, 1\}^{H \times W}, \tag{14}$$

where $H$ and $W$ denote the height and width of $I_t$. To enhance texture mask robustness via conventional detection [59], we adopt the Average Local Pixel Difference (ALPD) metric. Specifically, local and global texture metrics are computed as:

$$v_{\text{ALPD}} = \frac{1}{P} \sum_{i=1}^{P} |g_i - \bar{g}|, \quad v_{\text{global}} = \frac{w}{N_p} \sum_{j=1}^{N_p} |g_j - \bar{g}_{\text{global}}|, \tag{15}$$

where $g_i$ is the intensity of the $i$-th local pixel ($i = 1, \dots, P$), $\bar{g}$ is their mean, and $N_p$ is the pixel count in $I_t$. Scalar $w$ serves as a sensitivity coefficient. A pixel is classified as textured if $v_{\text{ALPD}} > v_{\text{global}}$. We then define a reweighted loss using this texture mask:

$$\mathcal{L}_{\text{texture-focus}} = \frac{1}{H \times W} \sum_{x,y} w(x,y) \cdot \rho(\hat{I}_t(x,y) - I_t(x,y)), \quad w(x,y) = 1 + \lambda M(x,y), \tag{16}$$

where $\lambda$ controls the emphasis on textured areas and $\rho(\cdot)$ denotes the Charbonnier loss. $\hat{I}_t(x,y)$ and $I_t(x,y)$ indicate the predicted and ground-truth frames, respectively. This formulation encourages more accurate reconstruction of real high-frequency details.

**Mask uncertainty penalty limitations.** Modern FI systems typically synthesize intermediate frames by fusing two consecutive inputs $I_0$ and $I_1$ using a dual-branch formulation:

$$\mathbf{F}^{(i)} = \mathbf{F}_0^{(i)} \odot \mathbf{M}^{(i)} + \mathbf{F}_1^{(i)} \odot (1 - \mathbf{M}^{(i)}), \tag{17}$$

where $\mathbf{F}_0^{(i)}$ and $\mathbf{F}_1^{(i)}$ is the $i$-th scale features of the two input frames, $\odot$ is element-wise multiplication, and $\mathbf{M}^{(i)} \in [0, 1]^{H_i \times W_i}$ is a learnable mask indicating the per-pixel dominance of each source. We posit that under ideal conditions where $\mathbf{M}^{(i)}$ should converge to binary values (0 or 1) for confident fusion, $\mathbf{M}^{(i)}$ around 0.5 often leads to ghosting or blur in synthetic frames. To address this, we introduce an uncertainty-aware regularization term applied to each mask level during training:

$$\mathcal{L}_{\text{mask}}^{(i)} = \lambda_i \cdot \frac{1}{H \times W} \sum_{p} \left[ 4 \cdot \mathbf{M}_p^{(i)} (1 - \mathbf{M}_p^{(i)}) \right]^{\gamma}, \quad 0 < \lambda_1 < \lambda_2 < \cdots < \lambda_S, \tag{18}$$

where $p$ traverses all pixel positions, $\gamma \in (0, 1]$ controls the penalty growth rate, and $\lambda_i (i = 1, 2, \cdots, S)$ is $i$-th scale-specific weight. Lower-resolution scales are allowed to retain higher uncertainty to better handle large motion and occlusion, while higher-resolution scales are encouraged to produce more confident, near-binary decisions by assigning larger $\lambda_i$. Additional formulation details are provided in Appendix A.4.

## 4 Experiments

### 4.1 Implementation

**Datasets for virtual camera generation.** We capture zoom image pairs from HuaweiPura70Ultra and RedmiK50Ultra [50] across 132 real-world scenes (79 indoor, 53 outdoor), covering diverse environments and illumination conditions. Based on Section 3.1, we generate 205 virtual sequences (16 frames each) for our ZI dataset. For ZI dataset construction details and representative samples, please refer to Appendix A.2 and A.3. We compare our ZI dataset with ZoomGS and 3DGS in terms of supervision quality and generalization ability for real-world ZI. Visual comparisons and finetune results are provided in Appendix A.5.

**Similarity map encoding computation.** During **training**, $t = Sim(i, j, k)$, $\forall (i, j, k)$. $Sim(i, j, k)$ is derived from ground-truth frames using RAFT and Lite-Mono (see Eq 13). During **inference**, the

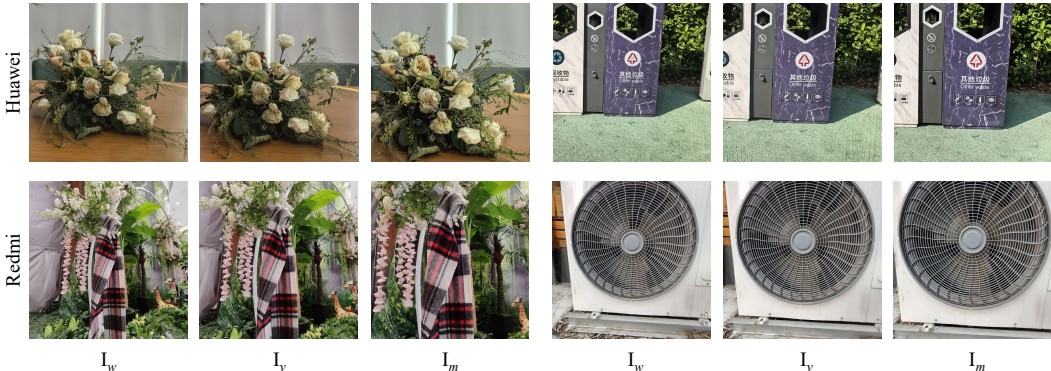

Figure 2: Sample triplets from our ZI dataset, each showing the wide-angle $I_w$, the synthetic intermediate frame $I_v$, and the main camera image $I_m$, for two devices: Huawei and Redmi.

ground-truth frame is not accessible, and the exact similarity map cannot be computed. Following the strategy used in prior work such as InterpAny [60], it is sufficient to provide a $t = fixed - timestep\ map$ in the same manner as conventional time-indexing methods. The semantics of this indexing map have shifted from an uncertain timestep map to a more deterministic motion hint. Physically, this encourages the model to move each object at constant speeds along their trajectories. In practice, this constant-speed assumption serves as a valid approximation for smooth zoom scenarios commonly found in mobile photography.

**Training configuration.** For virtual camera generation, we adopt the Adam optimizer [24] in a two-stage scheme. We first train the Gaussian model $\mathcal{M}_w$ for 5k iterations using the learning rate from 3DGS [23]. Then, joint optimization is performed for 30k iterations on both $\mathcal{M}_w$ and the ColorNet, with the learning rate decaying from 1e-3 to 1e-6 after 20k steps. Our 3D-TPR framework is integrated into RIFE [17], IFRNet [25], EMAVFI [53], and AMT [27]. Each model is initially trained on the Vimeo-90K [52] septuplet dataset (91,701 sequences at 448×256 resolution) using its default hyperparameters. We then finetune them on our ZI dataset. All experiments are implemented by PyTorch [35] on hardware devices equipped with 40 GB of memory.

**Evaluation configuration.** We evaluate performance and generalization on two types of datasets. For benchmarks with ground-truth, including Vimeo90K, Inter4K [41], and UCF101 [40], we use full-reference image quality metrics: PSNR [18], SSIM [49], and LPIPS [55]. For real-world evaluation, we collect zoom image pairs from four smartphone platforms: HuaweiPura70Ultra, iPhone16ProMax, OPPOFindX8Ultra, and RedmiK50Ultra[50]. Since ground truth is unavailable, we adopt no-reference metrics including NIQE [32], PI [1], CLIP-IQA [45], MUSIQ [22] and FLOLPIPS [11]. For all comparisons, the 2D baseline refers to the distance indexing paradigm [60]. For real-world deployment without geometric priors, the model uses a uniform map as motion guidance when inference. Real-world dataset details are provided in the Appendix A.7.

## 4.2 Ablation studies

To assess the effectiveness and compatibility of each component in our 3D-TPR framework, we perform a controlled ablation study on EMA-VFI in Table 1. The 2D framework [60] serves as the baseline. We evaluate four configurations: **(1) 3D-t-m**, which applies only the 3D-TPR encoding; **(2) 3D-m**, which integrates the texture-focus strategy; **(3) 3D-t**, which incorporates the mask penalty constraint; and **(4) 3D**, the full model with all components enabled.

Although improvements on pixel-centric metrics (PSNR and SSIM) appear modest, we argue that in most practical ZI scenarios, the objective is not to produce pixel-aligned frames, but to **synthesize perceptually plausible intermediate frames**. Pixel-centric metrics are relatively insensitive to common ZI artifacts such as blur and ghosting [56], which our design explicitly targets by **resolving velocity ambiguity**. As such, pixel-centric metrics are less informative for perceptual assessment. While the improvements on pixel-centric metrics (PSNR and SSIM) appear numerically modest, we emphasize that in practical Zoom Interpolation (ZI) scenarios, the objective is not to reproduce pixel-aligned frames but to **synthesize perceptually plausible intermediate frames**. Moreover, pixel-centric measures are inherently **insensitive to blur and ghosting artifacts**—the dominant

Table 1: Ablation study of the 3D-TPR framework. **Bold** values indicate the best results. Each component contributes incremental gains, and the full model yields the best performance.

| Method | PSNR ↑ | SSIM ↑ | LPIPS ↓ | PI ↓ | MUSIQ ↑ | FLOLPIPS ↓ |
|---|---|---|---|---|---|---|
| 2D | 24.73 | 0.851 | 0.081 | 5.071 | 55.968 | 0.128 |
| 3D-t-m | 24.78 | 0.852 | 0.081 | 5.041 | 56.615 | 0.119 |
| 3D-m | 24.74 | 0.852 | 0.084 | 5.022 | 57.362 | 0.113 |
| 3D-t | 24.80 | 0.853 | 0.081 | 4.934 | 56.946 | 0.116 |
| 3D (full) | **24.86** | **0.853** | **0.080** | **4.802** | **58.725** | **0.091** |

Table 2: Comparing the impact of optical flow on Vimeo90K.

| Benchmarks | Metrics | IFRNet | | | | AMT | | | |
|---|---|---|---|---|---|---|---|---|---|
| | | 1x | 2x | 4x | 8x | 1x | 2x | 4x | 8x |
| Vimeo90k | PSNR↑ | 27.21 | 27.21 | 26.01 | 25.01 | 27.25 | 27.22 | 26.01 | 24.94 |
| | SSIM↑ | 0.901 | 0.901 | 0.867 | 0.847 | 0.902 | 0.902 | 0.849 | 0.832 |
| | LPIPS↓ | 0.070 | 0.074 | 0.085 | 0.094 | 0.083 | 0.084 | 0.106 | 0.127 |

degradations caused by velocity ambiguity, which our framework is explicitly designed to mitigate. Consequently, these metrics are less indicative of perceptual fidelity.

In contrast, perceptual metrics such as PI, MUSIQ, and FLOLPIPS reveal consistent and significant improvements across all enhanced variants. Each component contributes distinct and complementary benefits: 3D-TPR encoding improves motion consistency (lower FLOLPIPS), texture-focus strategy enhances high-frequency details (higher MUSIQ), and mask uncertainty penalty suppresses occlusion artifacts (lower PI). When combined, the full model yield the best overall performance, confirming that our framework effectively enhances perceptual quality and motion coherence in ZI. Furthermore, when high-quality disparity supervision is available (Appendix A.10), the framework achieves notably larger gains, indicating its strong potential to exploit more accurate geometric priors.

To analyze the effect of optical flow accuracy on 3D-TPR, we downsample the RAFT inputs to 1×, 2×, 4×, and 8× resolutions. As RAFT is resolution-sensitive, this setup offers an effective proxy for flow fidelity variation. All estimated flows are bilinearly upsampled to original resolution for training on AMT and IFRNet. As shown in Table 2, lower-resolution flows (4×/8×) lead to noticeable degradation, while 2× setting achieves comparable performance to the 1x resolution case while significantly reducing computation. We therefore adopt 2× as our default configuration.

### 4.3 Qualitative results

To demonstrate the visual quality of our synthetic intermediate supervision, Figure 2 shows representative samples from the ZI dataset. $I_v$ achieves smooth transitions in photometrical and geometric structure, while faithfully preserving fine textures and high-frequency details. Despite being generated without ground-truth supervision, the results exhibit strong visual consistency and realism, providing effective guidance for training ZI models.

In Figure 3, we present visual comparisons between 2D and 3D-TPR methods on Vimeo90K. Across multiple FI networks, the 3D-TPR method consistently yields significantly better perceptual quality, producing clearer and sharper images. In contrast, 2D approaches often suffer from noticeable blur, while our 3D-TPR framework maintains fine-grained textures and structural consistency throughout. These results highlight the strong perceptual quality and robustness of the 3D-TPR framework.

Meanwhile, Figure 4 offers a qualitative comparison of ZI results across FI networks and devices. $3D_f$-TPR consistently produces higher-quality outputs with fewer artifacts, less blur, and sharper structures. Notably, RIFE, IFRNet, and AMT exhibit significantly reduced blur, while EMA-VFI produces clearer and more coherent line structures. Visual results demonstrate the effectiveness of our data generation method in cross-device generalization for the ZI task. Building upon this, OmniZoom ($3D_f$-TPR) further exhibits robust performance across diverse FI networks and mobile platforms, offering a universal solution for real-world ZI. Additional visual results are provided in Appendix A.8, and comparisons of 1D and 2D fine-tuning strategies are presented in Appendix A.9.

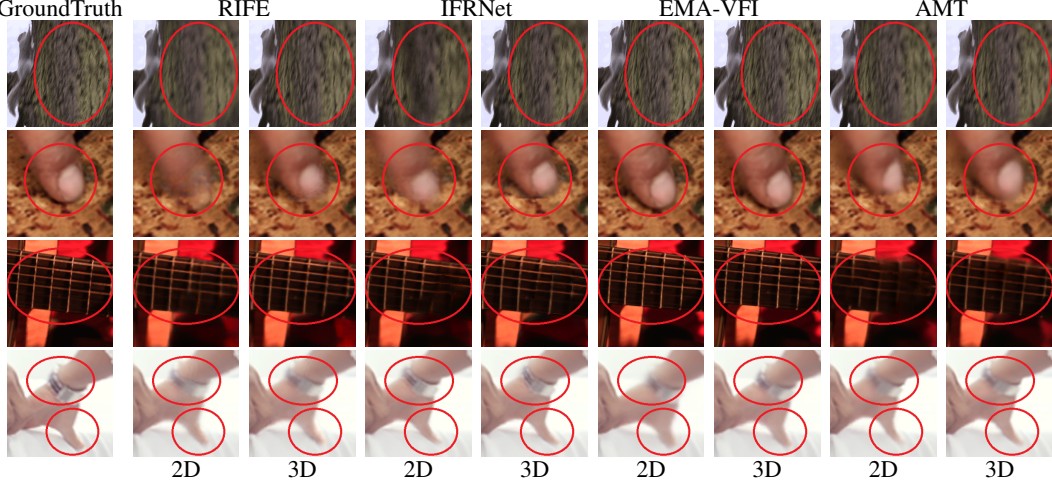

Figure 3: Visual comparison of 2D and 3D-TPR FI results across networks. Each row shows interpolation results at the same timestep.

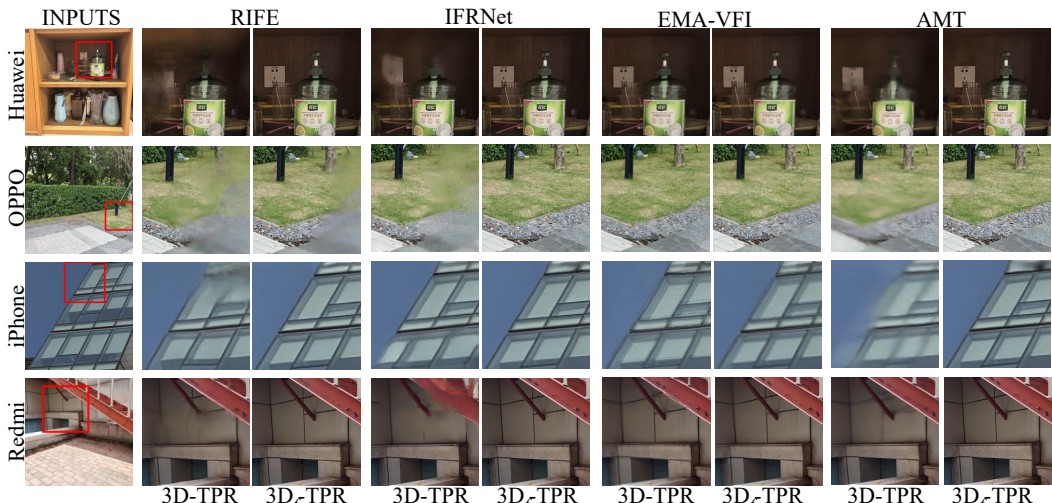

Figure 4: Qualitative results on real-world data across four FI networks at timestep $t = 1/2$. The subscript $_f$ denotes models finetuned on our ZI dataset.

## 4.4 Quantitative results

As shown in Table 3, we conduct quantitative comparisons between the 2D and 3D-TPR frameworks on Vimeo90K, Inter4K, and UCF101. On Vimeo90K, 3D-TPR achieves clear gains in PSNR, SSIM, and LPIPS, indicating better reconstruction and perceptual quality. On the unseen Inter4K and UCF101 benchmarks, 3D-TPR still improves perceptual metrics (SSIM and LPIPS), demonstrating strong generalization. Although pixel-centric metric PSNR may not always improve, ZI focuses on perceptually plausible synthesis rather than pixel-aligned outputs. Also, PSNR is less sensitive to blur [55], which is a common artifact in ZI. Combined with the visual results in Figure 3, it confirms the effectiveness and robustness of 3D-TPR for real-world ZI.

Table 4 offers quantitative results on real-world ZI tasks across four smartphone test sets under four FI networks. Due to space constraints, we use 3D in this table to represent our proposed 3D-TPR framework. 3D-TPR consistently achieves second-best performance across most devices and metrics, clearly outperforming the 2D baseline and demonstrating its effectiveness in handling ZI tasks even without domain-specific tuning. After finetuning on our ZI dataset, OmniZoom (3D$_f$-TPR) achieves the best results across nearly all models and devices, highlighting both the strong cross-device generalization capability of the dataset and the plug-and-play adaptability of the framework.

Table 3: Comparisons on Vimeo90k, Inter4K and UCF101 using 2D and 3D-TPR FI frameworks. **Bold** indicates the best metric.

| Benchmarks | Metrics | RIFE | | IFRNet | | EMA-VFI | | AMT | |
|---|---|---|---|---|---|---|---|---|---|
| | | 2D | 3D-TPR | 2D | 3D-TPR | 2D | 3D-TPR | 2D | 3D-TPR |
| Vimeo90k | PSNR↑ | 27.40 | **27.51** | 27.13 | **27.21** | 24.73 | **24.86** | 27.17 | **27.22** |
| | SSIM↑ | 0.901 | **0.902** | 0.899 | **0.901** | 0.851 | **0.853** | 0.902 | **0.902** |
| | LPIPS↓ | 0.086 | **0.081** | 0.078 | **0.074** | 0.081 | **0.080** | 0.081 | 0.084 |
| inter4K | PSNR↑ | 33.92 | **34.06** | **33.73** | 33.72 | 30.06 | **30.29** | 33.57 | **33.80** |
| | SSIM↑ | 0.951 | **0.952** | 0.952 | **0.952** | 0.903 | **0.904** | 0.953 | **0.955** |
| | LPIPS↓ | 0.048 | **0.046** | 0.046 | **0.045** | 0.044 | **0.044** | 0.048 | **0.047** |
| UCF101 | PSNR↑ | 35.54 | **35.85** | 35.42 | **35.43** | **36.74** | 36.65 | **35.37** | 35.33 |
| | SSIM↑ | 0.928 | **0.983** | 0.984 | **0.986** | 0.984 | **0.985** | 0.984 | **0.984** |
| | LPIPS↓ | **0.017** | 0.018 | 0.017 | **0.017** | 0.012 | **0.012** | 0.018 | **0.017** |

Table 4: Comparisons on real-world data across four FI models. Subscript $_f$ denotes models finetuned on our ZI dataset. **Bold** indicates the best performance, and underline is the second best.

| Device | Metrics | RIFE | | | IFRNet | | | EMA-VFI | | | AMT | | |
|---|---|---|---|---|---|---|---|---|---|---|---|---|---|
| | | 2D | 3D | $3D_f$ | 2D | 3D | $3D_f$ | 2D | 3D | $3D_f$ | 2D | 3D | $3D_f$ |
| Huawei | NIQE↓ | 3.8464 | _3.8651_ | **3.7422** | 3.6801 | **3.4852** | _3.5035_ | 3.6363 | _3.8470_ | 3.8341 | _3.5665_ | 3.6459 | **3.4547** |
| | PI↓ | 4.2505 | _4.1537_ | **3.9360** | 3.9570 | _3.3150_ | **3.2520** | **3.7240** | 3.8566 | _3.8467_ | _3.5147_ | 3.6132 | **3.3121** |
| | CLIP-IQA↑ | 0.3691 | _0.4939_ | **0.5233** | 0.5422 | _0.5784_ | **0.5909** | 0.5612 | _0.5621_ | **0.5684** | 0.5428 | _0.5498_ | **0.6018** |
| | MUSIQ↑ | 44.8268 | _58.8362_ | **60.8585** | 57.4632 | _73.0233_ | **74.1220** | 61.7263 | _62.7403_ | **63.5048** | 71.4369 | _71.7016_ | **73.7092** |
| iPhone | NIQE↓ | 4.2031 | _4.1027_ | **3.8601** | 3.8786 | **3.6923** | _3.6953_ | 3.8155 | _4.0555_ | 4.0934 | _3.5932_ | 3.7053 | **3.5087** |
| | PI↓ | 4.5340 | _4.3503_ | **4.0657** | 4.2090 | _3.3942_ | **3.3165** | 3.9994 | 4.1304 | _4.1123_ | _3.4293_ | 3.5367 | **3.2661** |
| | CLIP-IQA↑ | 0.4821 | _0.5366_ | **0.5492** | 0.5240 | _0.5829_ | **0.5949** | 0.5352 | _0.5387_ | **0.5503** | _0.5931_ | 0.5930 | **0.6211** |
| | MUSIQ↑ | 55.0577 | _59.6800_ | **60.7988** | 57.4088 | _73.3569_ | **73.8292** | 58.1422 | _58.6204_ | **60.1002** | 71.9805 | _72.1243_ | **73.6150** |
| OPPO | NIQE↓ | 4.6364 | _4.6100_ | **4.3968** | 4.5820 | 5.2039 | _5.1133_ | 4.5568 | _4.5166_ | **4.5071** | _4.9042_ | 5.3017 | **4.6699** |
| | PI↓ | 5.0848 | _5.0104_ | **4.6815** | 4.9749 | _4.9603_ | **4.7144** | 4.9569 | _4.9022_ | **4.8752** | _4.9676_ | 5.3808 | **4.5702** |
| | CLIP-IQA↑ | 0.4989 | _0.5363_ | **0.5830** | **0.5525** | 0.5212 | _0.5461_ | **0.5622** | 0.5590 | _0.5591_ | _0.4798_ | 0.4518 | **0.5595** |
| | MUSIQ↑ | 63.2645 | _65.7912_ | **67.1494** | 67.2635 | _75.2718_ | **75.4545** | 67.4327 | _67.5921_ | **67.6444** | _73.0842_ | 72.1563 | **75.2187** |
| Redmi | NIQE↓ | 5.0138 | _4.8764_ | **4.4720** | 5.0098 | _4.2837_ | **4.0695** | 4.4088 | _4.3424_ | **3.8852** | 5.1925 | _4.7426_ | **3.9835** |
| | PI↓ | 5.3615 | _5.0247_ | **4.5195** | 5.1108 | _3.5993_ | **3.3880** | 4.6708 | _4.5974_ | **4.5519** | 5.4335 | _4.0235_ | **3.3548** |
| | CLIP-IQA↑ | 0.4077 | _0.4664_ | **0.4930** | 0.4219 | _0.4913_ | **0.5152** | 0.4599 | _0.4807_ | **0.4947** | 0.4336 | _0.4754_ | **0.5383** |
| | MUSIQ↑ | 56.3870 | _61.9235_ | **63.6990** | 57.5453 | _73.7573_ | **74.3871** | 57.0371 | _59.6610_ | **60.7579** | 57.6671 | _71.9982_ | **74.3325** |

## 5 Conclusions

This paper tackles the ZI task that aims to synthesize geometrically and photometrically consistent intermediate frames between dual-camera images. We propose OmniZoom, a universal solution comprising two key components. First, we introduce a cross-device ZI dataset construction pipeline that enables device-agnostic supervision through spatial transition modeling, dynamic color adaptation, and cross-domain consistency learning. Second, we develop a plug-and-play 3D-TPR framework that enhances ZI accuracy and perceptual quality by leveraging 3D trajectory encoding and geometry-aware feature similarity. Experiments on public benchmarks and real-world test sets show that OmniZoom consistently improves performance across diverse FI models and devices. We believe this exploration bridges the gap between pre-trained interpolation models and real-world dual-camera systems, paving the way for more practical and adaptable ZI solutions.

## 6 Limitations

While OmniZoom is designed to be plug-and-play across various FI networks, achieving optimal performance may still require adjustments to the training schedule, such as tuning the learning rate warm-up strategy, rebalancing loss weights, or modifying batch size. These lightweight adjustments reflect the need for architecture-aware training when applying our framework to diverse settings.

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

# A   Appendix

**The content of the supplementary material involves:**

- Details of ColorNet in Appendix A.1.
- Dual-camera data sources for the ZI dataset in Appendix A.2.
- Details of the synthetic ZI dataset and computational cost in Appendix A.3.
- Formula reasoning of 3D-TPR framework in Appendix A.4.
- Comparative evaluation of virtual view synthesis method for ZI task in Appendix A.5.
- Additional experiment on disparity map sources in Appendix A.6.
- Details of the real-world multi-device ZI test sets in Appendix A.7.
- Additional visual results of OmniZoom on the real-world ZI test sets in Appendix A.8.
- Benchmarking ZI dataset across 1D, 2D, and 3D-TPR frameworks in Appendix A.9.
- Upper bound analysis of 3D-TPR framework in Appendix A.10.
- Additional visual results of 3D-TPR framework in Appendix A.11.
- Broader impacts in Appendix A.12.

**We provide a project page at** `https://omnizoom.github.io/OmniZoom/`**, where visual results are available, and the code/dataset will be released soon.**

## A.1   Details of ColorNet

### A.1.1   Architecture of ColorNet

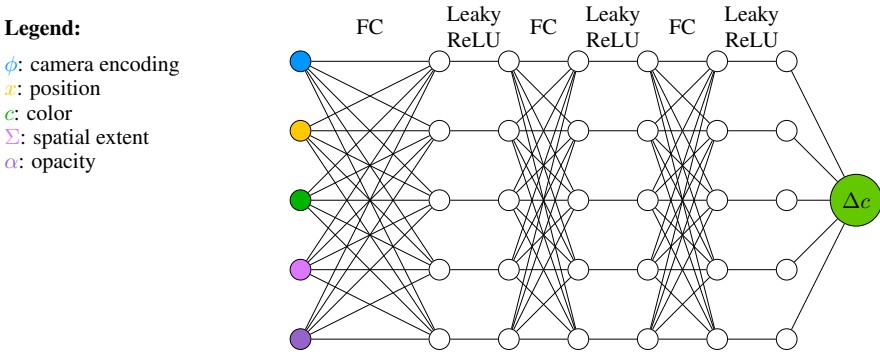

Figure 5: Architecture of ColorNet.

As illustrated in Figure 5, ColorNet is a lightweight multilayer perceptron (MLP) designed to estimate color residuals $\Delta c$ for neural 3D rendering. The network takes as input a five-dimensional feature tuple: the camera encoding $\phi$, the center position $x$, the base color $c$, the anisotropic covariance $\Sigma$, and the opacity $\alpha$. These inputs are first projected into a latent feature space through a fully connected (FC) layer, which is then processed by three consecutive FC blocks, each followed by a LeakyReLU activation to ensure non-linearity and mitigate the risk of neuron inactivation. Each FC layer contains 5 hidden units. Finally, the output layer maps the intermediate representation to a 3-dimensional color offset vector $\Delta c$ via a linear transformation. This residual is then added to the original color $c$, yielding the refined output color used for rendering.

### A.1.2   Ablations of ColorNet

We perform a no-reference image quality comparison between our proposed ColorNet and several common photometric calibration baselines: (1) *No Calibration*, (2) *Mean-Std Normalization* [48], and (3) *Reinhard Color Transfer* [51]. The evaluations are conducted on rendered intermediate frames using four widely adopted metrics: NIQE, PI, CLIP-IQA, and MUSIQ.

Table 5: No-reference image quality comparison of ColorNet against common photometric calibration baselines on rendered intermediate frames. Lower NIQE and PI indicate better perceptual quality, while higher CLIP-IQA and MUSIQ denote better alignment with human perception. ColorNet consistently achieves the best performance across all metrics, demonstrating its effectiveness in modeling fine-grained photometric variations.

| Method | NIQE ↓ | PI ↓ | CLIP-IQA ↑ | MUSIQ ↑ |
|---|---|---|---|---|
| No Calibration | 3.5832 | 3.8378 | 0.5193 | 68.4036 |
| Mean-Std Norm | 3.3193 | 3.7757 | 0.4577 | 63.0220 |
| Reinhard Transfer | 3.2357 | 3.7345 | 0.4588 | 64.3218 |
| **ColorNet (Ours)** | **2.8885** | **3.1515** | **0.6099** | **69.3328** |

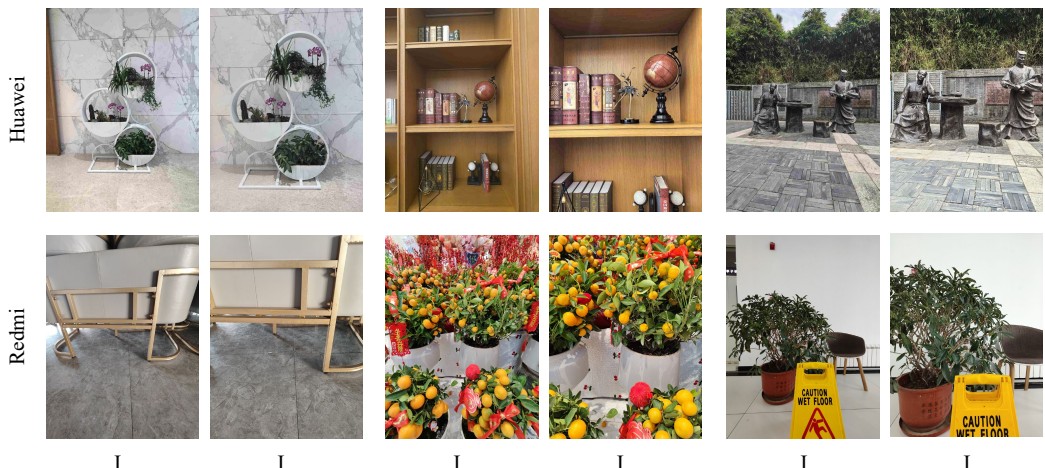

$I_w$       $I_m$       $I_w$       $I_m$       $I_w$       $I_m$

Figure 6: Representative dual-camera data sources. The top row displays samples from HuaweiPura70Ultra, while the bottom row presents examples from the publicly available RedmiK50Ultra dataset [50]. Each scene contains paired wide-angle ($I_w$) and main-camera ($I_m$) images, covering diverse content and geometric configurations.

As reported in Table 5, **ColorNet consistently outperforms all baseline methods**, demonstrating its superior capability in capturing fine-grained photometric variations that simpler approaches fail to model. These results highlight the effectiveness of ColorNet in enhancing color consistency and perceptual quality across intermediate frames.

## A.2    Dual-camera data sources for the ZI dataset

As mentioned in the main paper, our ZI dataset comprises dual-camera zoom image pairs from HuaweiPura70Ultra and RedmiK50Ultra. Specifically, we collect Huawei data ourselves across a variety of indoor and outdoor scenarios, while the Redmi sequences are adopted from the publicly available dataset introduced in [50]. The image resolutions vary by device: 2133×1600 for Huawei and 1632×1224 for Redmi, consistent with their native imaging pipelines.

In total, the combined dataset covers 132 real-world scenes, consisting of 79 indoor and 53 outdoor environments. Indoor scenes include structured environments such as classrooms, shopping malls, and cafeterias, while outdoor scenes cover playgrounds, parks, and other open spaces with greater geometric variation. For each scene, we capture 15 pairs of main–wide camera images, along with 6 additional wide-angle views from peripheral viewpoints to enhance spatial diversity. These supplementary images increase the geometric baseline and play a key role in stabilizing 3DGS optimization. Representative examples from both devices are shown in Figure 6.

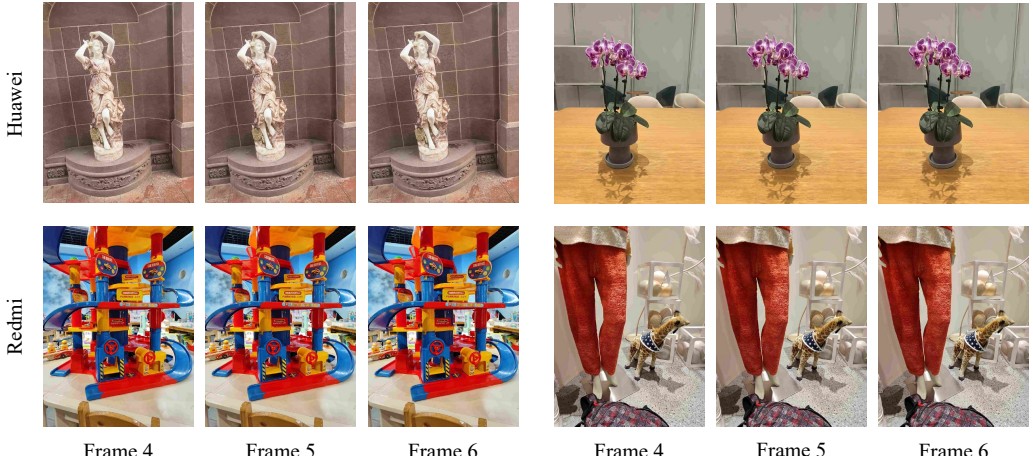

|      | Frame 4 | Frame 5 | Frame 6 | Frame 4 | Frame 5 | Frame 6 |

Figure 7: Representative frames (Frames 4, 5, and 6) from our synthetic ZI dataset. Each row illustrates three temporally adjacent frames sampled from a 16-frame virtual zoom sequence, demonstrating smooth transitions in both appearance and geometry. Examples are shown from Huawei and Redmi devices, covering diverse indoor and outdoor scenes with varying structural complexity.

### A.3 Details of the synthetic ZI dataset and computational cost

We generate 205 synthetic zoom sequences, each consisting of 16 interpolated frames. These sequences serve as high-quality supervision for training and evaluation in ZI tasks. Representative examples of the rendered dataset are shown in Figure 7.

The training duration for each scene depends on its visual complexity and structural content, with an average of approximately 1.5 hours per scene. Rendering a complete zoom sequence with 16 interpolated frames typically takes less than 20 seconds. All procedures are conducted on a workstation with 40 GB of memory, consistent with our main experimental setup.

### A.4 Formula reasoning of 3D-TPR framework

#### A.4.1 Mathematical modeling and depth constraints for projective ambiguity

**Modeling the projection from 3D space to 2D image plane**: Let $\mathbf{P} = (X, Y, Z)^T$ denote a 3D point in the scene, which is projected onto 2D image coordinates $(x, y)$ via the pinhole camera model. The projection is formulated as:

$$x = \frac{fX}{Z}, \quad y = \frac{fY}{Z}; \quad \pi(\mathbf{P}) = \left(\frac{fX}{Z}, \frac{fY}{Z}\right)^T, \tag{19}$$

where $\pi$ denotes the 3D-to-2D projection function, and $f$ represents the camera's focal length. When an object moves through 3D space, its instantaneous displacement over a time interval $\Delta t$ is governed by its 3D velocity vector $\mathbf{v} = (v_X, v_Y, v_Z)^T$:

$$\Delta \mathbf{P} = \mathbf{v}\Delta t = (v_X \Delta t, v_Y \Delta t, v_Z \Delta t)^T, \tag{20}$$

where $\Delta \mathbf{P}$ denotes the change in 3D position during $\Delta t$. Let $I(\mathbf{P}, t)$ denote the projection of a 3D point $\mathbf{P} = (X, Y, Z)^T$ onto the image plane at time $t$. When the object is in motion, the position of $\mathbf{P}$ evolves over time as:

$$\mathbf{P}(t + \Delta t) = \mathbf{P}(t) + \Delta \mathbf{P} = \mathbf{P}(t) + \mathbf{v}\Delta t. \tag{21}$$

The corresponding image plane displacement $\delta = (\Delta x, \Delta y)$ induced by 3D motion can be approximated via a first-order Taylor expansion:

$$\delta = \pi(\mathbf{P} + \mathbf{v}\Delta t) - \pi(\mathbf{P}) \approx \frac{\partial \pi}{\partial \mathbf{P}}\mathbf{v}\Delta t, \tag{22}$$

where $\frac{\partial \pi}{\partial \mathbf{P}}$ is the projection Jacobian matrix $J_\pi$, which characterizes how variations in 3D positions affect the projected 2D coordinates. The term $\mathbf{v}\Delta t$ represents the instantaneous 3D displacement, and its transformation through $J_\pi$ yields the corresponding image plane motion. By substituting the partial derivatives of the projection function, the Jacobian matrix $J_\pi$ can be explicitly written as:

$$J_\pi = \begin{bmatrix} \frac{\partial x}{\partial X} & \frac{\partial x}{\partial Y} & \frac{\partial x}{\partial Z} \\ \frac{\partial y}{\partial X} & \frac{\partial y}{\partial Y} & \frac{\partial y}{\partial Z} \end{bmatrix} = \begin{bmatrix} \frac{f}{Z} & 0 & -\frac{fX}{Z^2} \\ 0 & \frac{f}{Z} & -\frac{fY}{Z^2} \end{bmatrix}. \tag{23}$$

Equation 23 constitutes an underdetermined system of equations, whose solution space forms a straight line. Mathematically, this can be expressed as:

$$\mathbf{v} = \mathbf{v}_0 + k\mathbf{n}, \quad \mathbf{n} = \left( \frac{X}{Z}, \frac{Y}{Z}, 1 \right)^T, \tag{24}$$

where $\mathbf{v}_0$ represents a particular solution, $\mathbf{n}$ denotes the direction corresponding to the viewing ray, and $k \in \mathbb{R}$ denotes an arbitrary real number. This formulation implies that infinitely many 3D velocity vectors, which differ only in their components along the line of sight, can result in the same 2D image plane displacement. As a concrete example, consider a 3D scene point located at $\mathbf{P} = (f, 0, Z)^T$:

**Scene 1.** Suppose the object moves purely along the horizontal axis with the velocity vector $\mathbf{v}_1 = (1, 0, 0)^T$. The resulting image plane displacement $\delta_1$ can be approximated via the projection Jacobian $J_\pi$ as:

$$\delta_1 = J_\pi \cdot \mathbf{v}_1 \cdot \Delta t = \begin{bmatrix} \frac{f}{Z} \cdot 1 + 0 \cdot 0 + \left( -\frac{fX}{Z^2} \right) \cdot 0 \\ 0 \cdot 1 + \frac{f}{Z} \cdot 0 + \left( -\frac{fY}{Z^2} \right) \cdot 0 \end{bmatrix} \cdot \Delta t = \begin{bmatrix} \frac{f}{Z} \\ 0 \end{bmatrix} \cdot \Delta t. \tag{25}$$

**Scene 2.** Now consider the case where the object moves along the optical axis (i.e., in depth), with the velocity vector $\mathbf{v}_2 = (0, 0, -Z/f)^T$. For the 3D point $\mathbf{P} = (f, 0, Z)^T$, the resulting image plane displacement $\delta_2$ is computed as:

$$\delta_2 = J_\pi \cdot \mathbf{v}_2 \cdot \Delta t = \begin{bmatrix} \frac{f}{Z} \cdot 0 + 0 \cdot 0 + \left( -\frac{fX}{Z^2} \right) \cdot \left( -\frac{Z}{f} \right) \\ 0 \cdot 0 + \frac{f}{Z} \cdot 0 + \left( -\frac{fY}{Z^2} \right) \cdot \left( -\frac{Z}{f} \right) \end{bmatrix} \cdot \Delta t = \begin{bmatrix} \frac{X}{Z} \\ \frac{Y}{Z} \end{bmatrix} \cdot \Delta t = \begin{bmatrix} \frac{f}{Z} \\ 0 \end{bmatrix} \cdot \Delta t. \tag{26}$$

The distinct 3D motions illustrated in **Scene 1** and **Scene 2** become indistinguishable when projected onto the 2D image plane, thereby *leading to the non-uniqueness of motion estimation* and *highlighting the necessity of depth-direction constraints*. By enforcing a constraint along the depth direction, the solution space is reduced from a continuous line to a single, physically plausible point, effectively eliminating the ambiguity and suppressing implausible variations along the depth axis.

### A.4.2 Gradient reallocation with texture-focus strategy

Let $\mathcal{S}_T = (x, y) \mid \mathbf{M}(x, y) = 1$ denote the set of texture pixels, where $\mathbf{M}$ is the binary mask defined in Equation 14, and let $\mathcal{S}_C$ denote its complement. According to Equation 16, the expected value of gradient norm over texture regions satisfies the following for $\lambda > 0$:

$$\mathbb{E}\left[ \|\nabla_\theta \mathcal{L}_{texture-focus}\|_{\mathcal{S}_T} \right] = (1 + \lambda) \cdot \mathbb{E}\left[ \|\nabla_\theta \mathcal{L}_{base}\|_{\mathcal{S}_T} \right]. \tag{27}$$

This formulation indicates that the texture-focus loss reweights the gradient flow to amplify supervision in structurally informative regions. By scaling the gradient magnitude in $\mathcal{S}_T$ by a factor of $(1 + \lambda)$, the network is encouraged to prioritize learning from texture-rich areas, which are typically more perceptually salient and semantically meaningful.

While expected value of the gradient norm within the texture region $\mathcal{S}_T$ is magnified by a factor of $(1 + \lambda)$, the gradient norm in the non-texture region $\mathcal{S}_C$ remains unchanged:

$$\mathbb{E}\left[ \|\nabla_\theta \mathcal{L}_{\text{texture-focus}}\|_{\mathcal{S}_C} \right] = \mathbb{E}\left[ \|\nabla_\theta \mathcal{L}_{\text{base}}\|_{\mathcal{S}_C} \right]. \tag{28}$$

Consequently, the texture-focus strategy effectively redistributes the gradient energy in the parameter space by selectively amplifying directions associated with texture regions, thereby enhancing supervision in structurally informative areas. Under momentum-based stochastic gradient descent (SGD), the network parameters are updated as:

$$\theta_{k+1} = \theta_k - \eta \mathbf{v}_k, \quad \mathbf{v}_k = \beta \mathbf{v}_{k-1} + (1 - \beta) \nabla_\theta \mathcal{L}_{\text{texture-focus}}(\theta_k), \tag{29}$$

where $\theta_k$ denotes the network parameters at the $k$-th iteration, and $\theta_{k+1}$ represents the updated parameters after applying a weighted update in the direction of the momentum term $\mathbf{v}_k$. Here, $\eta$ is the learning rate, and $\mathbf{v}_k$ is the accumulated momentum, which integrates the current gradient with past updates. We set the momentum coefficient to $\beta = 0.9$, which determines the proportion of the previous momentum $\mathbf{v}_{k-1}$ retained in the current step. The gradient term $\nabla_\theta \mathcal{L}_{texture-focus}(\theta_k)$ is computed with respect to the texture-focus loss at iteration $k$.

Since the gradient of $\mathcal{L}_{texture-focus}$ within the texture region $\mathcal{S}_T$ is scaled by a factor of $(1 + \lambda)$, the corresponding entries in the gradient tensor exhibit proportionally larger magnitudes. This induces an optimization bias toward texture-dense regions, thereby encouraging edge sharpening and the recovery of high-frequency details. In contrast, gradients within the non-texture region $\mathcal{S}_C$ remain unchanged. This targeted reallocation strategy prioritizes learning in structurally salient areas, ultimately enhancing the perceptual fidelity and fine-detail quality of the interpolated frames.

Our proposition formally demonstrates that explicitly amplifying momentum components along high-frequency texture directions offers two key benefits: (a) it reduces the number of iterations required to reach local optima by modulating the effective learning rate, and (b) it suppresses oscillatory behaviors in the loss surface caused by the competing objectives of texture underfitting and background overfitting.

Under the flat-minima hypothesis for over-parameterized networks, this gradient reallocation strategy further flattens the loss landscape during finetuning, acting as an implicit regularizer that improves generalization to unseen texture distributions through curvature-driven optimization dynamics. Unlike perceptual losses that entangle high- and low-frequency signals within the shared feature space of pretrained encoders, our texture-focus strategy achieves frequency disentanglement via pixel-level gradient modulation in an encoder-agnostic manner, allowing more precise control over structural detail reconstruction only with an ancillary supervision.

### A.4.3 Additional proof of mask penalty strategy

Normalization coefficient $4 \cdot \mathbf{M}(1 - \mathbf{M}) \in [0, 1]$, reaching the maximum peak value at $\mathbf{M} = 0.5$. As $\mathbf{M} \to 0$ or $\to 1$, the normalization coefficient value tends to zero. From a Bayesian perspective, the mask value $\mathbf{M}_p^{(i)}$ is modelled as the parameter $\theta_p^{(i)}$ of the posterior Bernoulli distribution $\mathcal{Q}$:

$$(\mathbf{M}_p^{(i)} = \theta_p^{(i)}) \sim (\mathcal{Q} = \mathbf{Bernoulli}(\theta_p^{(i)})), \tag{30}$$

where $\mathbf{M}_p^{(i)}$ represents the model-inferred confidence at pixel position $p$ of the $i$-th scale, which is calculated by the middle layer of the network and equivalent to $\theta_p^{(i)}$. $\theta_p^{(i)}$ is the parameter of the Bernoulli distribution, representing the probability that the mask value is 1 at position $p$, indicating the network's confidence that the current pixel is dominated by the features from $I_0$.

The case $\theta_p^{(i)} = 1$ indicates that the mask value at position $p$ is fully dominated by $I_0$, whereas $\theta_p^{(i)} = 0$ implies complete dominance by $I_1$. $\theta_p^{(i)} = 0.5$ shares the equal contributions from both two reference frames.

The interpolation network dynamically adjusts the weights by learning the distribution of $\theta_p^{(i)}$. Since $\theta_p^{(i)}$ is directly adopted as the parameter of the Bernoulli distribution, its entropy $\mathcal{H}(\theta_p^{(i)})$ quantifies the uncertainty in the fusion decision:

$$\mathcal{H}(\mathcal{Q}) = \mathcal{H}(\theta_p^{(i)}) = -\theta_p^{(i)} \log \theta_p^{(i)} - (1 - \theta_p^{(i)}) \log(1 - \theta_p^{(i)}). \tag{31}$$

A higher entropy value indicates greater uncertainty in the network's fusion decision at the current pixel, typically when $\theta_p^{(i)}$ approaches 0.5. In contrast, lower entropy represents higher confidence, corresponding to $\theta_p^{(i)}$ values close to 0 or 1. Assume that an implicit Beta prior distribution $\mathcal{P}$ is applied to $\theta_p^{(i)}$:

$$\mathcal{P}(\theta_p^{(i)}) \propto \theta_p^{\alpha-1}(1 - \theta_p)^{\beta-1}, \tag{32}$$

where we adopt a symmetric, U-shaped configuration by setting $\alpha = \beta < 1$. The probability density is higher at $\theta_p = 0$ or $\theta_p = 1$ and lower at median $\theta_p = 0.5$, thereby encoding prior knowledge that induces a binarization tendency. Minimizing the KL divergence between the posterior distribution $\mathcal{Q}$ and the prior distribution $\mathcal{P}$ yields:

$$D_{KL}(\mathcal{Q}||\mathcal{P}) = \mathcal{H}(\mathcal{Q}, \mathcal{P}) - \mathcal{H}(\mathcal{Q}), \tag{33}$$

where $\mathcal{H}(\mathcal{Q}, \mathcal{P}) = -\mathbb{E}_{\mathcal{Q}}[\log \mathcal{P}(\theta_p^{(i)})]$ measures the cross-entropy between the posterior distribution $\mathcal{Q}$ and the prior distribution $\mathcal{P}$, reflecting how well the posterior aligns with the prior. A smaller value indicates that $\mathcal{Q}$ aligns more closely with the distributional preferences encoded in $\mathcal{P}$. $\mathcal{H}(\mathcal{Q})$ quantifies the intrinsic uncertainty of $\mathcal{Q}$, with a higher value indicating increased ambiguity in decision-making.

By designing $\mathcal{L}_{mask}^{(i)} \propto \theta_p^{(i)}(1 - \theta_p^{(i)})$, as derived in Equation 18, we implicitly impose a gradient direction consistent with the low-entropy Beta prior distribution $\mathcal{P}$. When $\theta_p^{(i)}$ approaches to $0.5$, the gradient of $\mathcal{H}(\mathcal{Q}, \mathcal{P})$ dominates the optimization direction since $\log \mathcal{P}(\theta_p^{(i)})$ tends to be negative infinity. In that case, $\theta_p^{(i)}$ will be forced to flee quickly from high-uncertainty areas, in alignment with the gradient of $\mathcal{L}_{mask}^{(i)}$. Once $\theta_p^{(i)}$ enters the vicinity of 0 or 1, the gradient of $\mathcal{H}(\mathcal{Q}, \mathcal{P})$ decays, and the gradient of $\mathcal{H}(\mathcal{Q})$ begins to dominate. At this point, the gradient of $\mathcal{L}_{mask}^{(i)}$ remains persistently operative throughout the optimization process, thus ensuring stable convergence of the model. Consequently, minimizing $\mathcal{L}_{mask}^{(i)}$ effectively approximates the minimizing KL divergence $D_{KL}(\mathcal{Q}||\mathcal{P})$. This design is theoretically anchored in the Bayesian framework and promotes training stability through a principled simplification of the target loss.

To sum up, the overall training target loss is:

$$\mathcal{L}_{overall} = \alpha_T \cdot \mathcal{L}_{texture-focus} + \alpha_m \cdot \sum_{i=1}^{S} \mathcal{L}_{mask}^{(i)} + \alpha_o \cdot \mathcal{L}_{original}, \tag{34}$$

where $\alpha T$, $\alpha_m$, and $\alpha_o$ denote the weighting coefficients for the texture-focus loss, mask regularization loss, and baseline supervision loss, respectively. These weights can be manually specified or adaptively optimized during training, depending on the architectural design and optimization dynamics of the model.

## A.5 Comparative evaluation of virtual view synthesis method for ZI task

### A.5.1 Comparison of virtual frame generation quality

To evaluate the effectiveness of synthetic supervision for the ZI task, we compare three virtual frame generation methods: ZoomGS [50], 3DGS [23], and our proposed pipeline. For a fair comparison, all baselines are reproduced using their official implementations and default hyperparameters. As shown in Figure 8, our method yields noticeably higher-quality intermediate views, exhibiting sharper textures, reduced artifacts, and more accurate geometric consistency. In contrast, ZoomGS and 3DGS often exhibit geometric distortion and texture degradation, particularly in scenes with complex spatial layouts or high-frequency surface details. These findings highlight the superiority of our pipeline in producing visually reliable and semantically aligned supervision tailored for ZI training.

### A.5.2 Evaluating virtual supervision via ZI finetuning

To assess the effectiveness of our synthetic supervision in real-world ZI scenarios, we conduct a downstream finetuning evaluation against two baselines: ZoomGS [50] and 3DGS [23]. For each method, the corresponding virtual dataset is used to finetune four representative FI networks, all integrated into our unified 3D-TPR framework. The resulting models are then evaluated on real-world test sets captured from four distinct smartphone platforms, enabling a comprehensive assessment of cross-device generalization performance.

**Quantitative results.** Table 6 reports results on Huawei and Redmi devices, both of which are included in all three synthetic ZI datasets. Our method consistently outperforms ZoomGS and 3DGS across all networks and metrics, demonstrating strong supervision quality in seen-device scenarios. While ZoomGS performs reasonably due to its device-specific training coverage, it lacks explicit spatial and color modeling. In contrast, our spatial transition and dynamic color adaptation modules effectively handle cross-camera misalignment and photometric variation, resulting in sharper textures and more coherent frame synthesis, reflected by gains in CLIP-IQA and MUSIQ. 3DGS also employs Huawei and Redmi data, but builds on monocular reconstruction assumptions and ignores dual-camera geometric or ISP discrepancies. Consequently, the composite supervision shows less consistency, which universally degrades finetuning efficacy on all tasks.

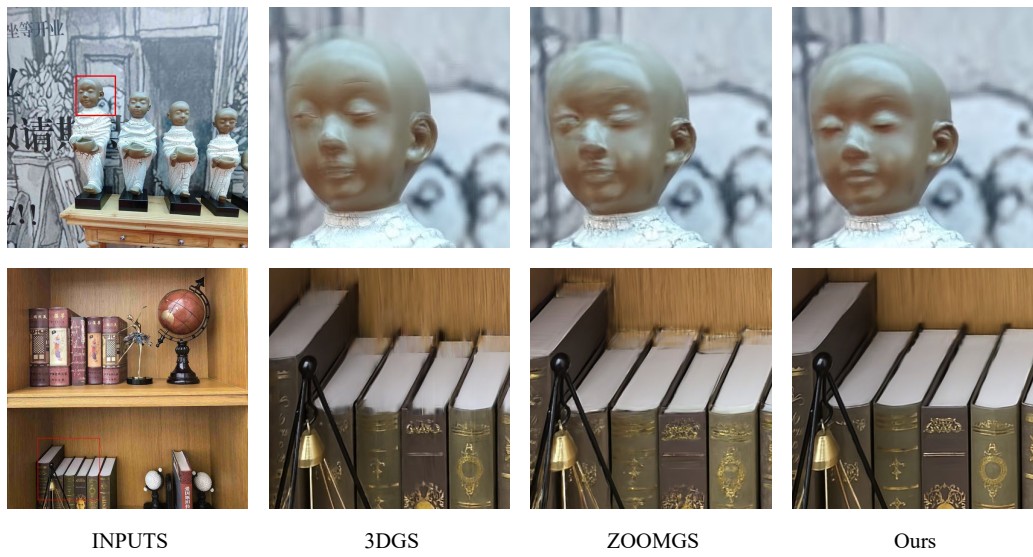

| INPUTS | 3DGS | ZOOMGS | Ours |

Figure 8: Visual comparison of virtual frame generation methods for ZI supervision. Compared to ZoomGS [50] and 3DGS [23], our method produces intermediate frames with fewer artifacts and better detail preservation, demonstrating superior supervision quality for training ZI models.

Table 6: Quantitative comparison on real-world dual-camera test sets captured by Huawei and Redmi devices, which are included in the synthetic dataset construction. Each FI model is evaluated under four supervision variants: original 3D-TPR pretrained (Base), and finetuned on 3DGS, ZoomGS, and our proposed supervision $_f$. **Bold** indicates the best result, and underline denotes the second best.

| Networks | Methods | HuaweiPura70Pro | | | | RedmiK50Ultra | | | |
| --- | --- | --- | --- | --- | --- | --- | --- | --- | --- |
| | | NIQE↓ | PI↓ | CLIP-IQA↑ | MUSIQ↑ | NIQE↓ | PI↓ | CLIP-IQA↑ | MUSIQ↑ |
| RIFE | Base | 3.8651 | 4.1537 | 0.4939 | 58.8362 | 4.8764 | 5.0247 | 0.4664 | 61.9235 |
| | 3DGS$_f$ | 3.8309 | 4.2570 | 0.5183 | 60.2724 | 4.8457 | 5.1787 | 0.3459 | 52.4562 |
| | ZoomGS$_f$ | 3.7902 | 4.0084 | 0.5013 | 59.0780 | 4.5598 | 4.7340 | 0.4716 | 61.2694 |
| | Ours$_f$ | **3.7422** | **3.9360** | **0.5233** | **60.8585** | **4.4720** | **4.5195** | **0.4930** | **63.6990** |
| IFRNet | Base | **3.4852** | 3.3150 | 0.5784 | 73.0233 | 4.2837 | 3.5993 | 0.4913 | 73.7573 |
| | 3DGS$_f$ | 3.5310 | 3.2570 | 0.5183 | 70.2725 | 4.2232 | 3.5776 | 0.4259 | 70.0659 |
| | ZoomGS$_f$ | 3.5132 | 3.2769 | 0.5867 | 74.0218 | 4.0879 | 3.3928 | 0.5127 | 74.2312 |
| | Ours$_f$ | 3.5035 | **3.2520** | **0.5909** | **74.1220** | **4.0695** | **3.3880** | **0.5152** | **74.3871** |
| EMA-VFI | Base | 3.8470 | 3.8566 | 0.5621 | 62.7403 | 4.3424 | 4.5974 | 0.4807 | 59.6610 |
| | 3DGS$_f$ | 3.8888 | **3.4184** | 0.5540 | 62.7141 | 4.2464 | **4.4593** | 0.4908 | 59.2312 |
| | ZoomGS$_f$ | **3.6360** | 3.8499 | 0.5229 | 61.2508 | 4.2106 | 4.4610 | 0.4718 | 60.1914 |
| | Ours$_f$ | 3.8341 | 3.8467 | **0.5684** | **63.5048** | **3.8852** | 4.5519 | **0.4947** | **60.7579** |
| AMT | Base | 3.6459 | 3.6132 | 0.5498 | 71.7016 | 4.7426 | 4.0235 | 0.4754 | 71.9982 |
| | 3DGS$_f$ | 3.4894 | 3.3929 | 0.5184 | 70.5856 | 4.3934 | 3.7834 | 0.4646 | 71.4254 |
| | ZoomGS$_f$ | 3.4593 | 3.3395 | 0.5847 | 73.7058 | 4.0653 | 3.4472 | 0.5209 | **74.4339** |
| | Ours$_f$ | **3.4547** | **3.3121** | **0.6018** | **73.7092** | **3.9835** | **3.3548** | **0.5383** | 74.3325 |

Table 7 evaluates generalization to iPhone and OPPO: two unseen devices that are not involved in training or synthetic data generation. Despite the domain gap, our method consistently achieves the best performance across all networks and evaluation metrics. In contrast, ZoomGS and 3DGS exhibit clear performance degradation due to their limited capacity to model cross-device geometric and

photometric variations. These results highlight the strong generalization ability of our pipeline, which provides robust, device-agnostic supervision across heterogeneous smartphone platforms.

Our method consistently outperforms existing virtual frame generation pipelines across both seen and unseen device scenarios. These gains are attributed to the explicit modeling of cross-device geometric

Table 7: Quantitative evaluation of cross-device generalization on iPhone and OPPO, two unseen devices excluded from the synthetic dataset generation. Despite the domain shift, models finetuned with our supervision consistently outperform those trained with 3DGS and ZoomGS. **Bold** indicates the best result, and underline denotes the second best.

| Networks | Methods | iPhone16ProMax | | | | OPPOFindX8Ultra | | | |
| --- | --- | --- | --- | --- | --- | --- | --- | --- | --- |
| | | NIQE↓ | PI↓ | CLIP-IQA↑ | MUSIQ↑ | NIQE↓ | PI↓ | CLIP-IQA↑ | MUSIQ↑ |
| RIFE | Base | 4.1027 | 4.3503 | 0.5366 | 59.6800 | 4.6100 | 5.0104 | 0.5366 | 65.7912 |
| | 3DGS$_f$ | 3.9618 | 4.2914 | 0.4687 | 54.5013 | 4.4927 | 4.9957 | 0.4408 | 60.6637 |
| | ZoomGS$_f$ | 3.8950 | 4.1679 | 0.5411 | 59.6339 | 4.4543 | 4.8111 | 0.5779 | 66.6404 |
| | Ours$_f$ | **3.8601** | **4.0657** | **0.5492** | **60.7988** | **4.3968** | **4.6815** | **0.5830** | **67.1494** |
| IFRNet | Base | 3.6923 | 3.3942 | 0.5829 | 73.3569 | 5.2039 | 4.9603 | 0.5212 | 75.2718 |
| | 3DGS$_f$ | **3.6233** | 3.3400 | 0.5583 | 71.7621 | 5.1306 | 4.8809 | 0.4755 | 74.3894 |
| | ZoomGS$_f$ | 3.7501 | 3.3498 | 0.5941 | 73.7598 | 5.1630 | 4.7927 | 0.5364 | 75.3461 |
| | Ours$_f$ | 3.6953 | **3.3165** | **0.5949** | **73.8292** | **5.1133** | **4.7144** | **0.5461** | **75.4545** |
| EMA-VFI | Base | 4.0555 | 4.1304 | 0.5387 | 58.6204 | 4.5166 | 4.9022 | 0.5590 | 67.5921 |
| | 3DGS$_f$ | **3.6258** | **3.8183** | 0.5418 | 59.2020 | 4.8225 | 4.9720 | **0.5868** | 66.8282 |
| | ZoomGS$_f$ | 3.8068 | 3.9542 | 0.5377 | 59.5872 | 4.5455 | 4.8818 | 0.5551 | **68.2806** |
| | Ours$_f$ | 4.0934 | 4.1123 | **0.5503** | **60.1002** | **4.5071** | 4.8752 | 0.5591 | 67.6444 |
| AMT | Base | 3.7053 | 3.5367 | 0.5930 | 72.1243 | 5.3017 | 5.3808 | 0.4518 | 72.1563 |
| | 3DGS$_f$ | 3.6991 | 3.4205 | 0.5644 | 71.6435 | 4.9154 | 5.0089 | 0.4639 | 72.5701 |
| | ZoomGS$_f$ | 3.5192 | 3.4044 | 0.5985 | 71.8655 | 4.6889 | 4.7544 | 0.5581 | 72.6175 |
| | Ours$_f$ | **3.5087** | **3.2661** | **0.6211** | **73.6150** | **4.6699** | **4.5702** | **0.5595** | **75.2187** |

transitions and photometric inconsistencies, two key factors for ensuring structurally accurate and perceptually consistent zoom interpolation in real-world.

**Qualitative results.** We further perform a qualitative comparison of interpolation outcomes across four FI networks and four smartphone platforms in real-world ZI tasks. As shown in Figure 9, we visualize the results of models finetuned with three types of synthetic supervision: ZoomGS, 3DGS, and our proposed approach, alongside the original 3D-TPR pretrained baselines without finetuning. ZoomGS$_f$ performs reasonably on Huawei and Redmi devices, which are included in its training dataset. However, it still suffers from noticeable blurring and diminished structural fidelity, particularly around object boundaries and fine textures. 3DGS$_f$, which lacks explicit device-specific modeling, exhibits limited generalization and introduces geometric distortions and color artifacts across all devices. In contrast, our method consistently generates sharper structures, more coherent textures, and significantly fewer artifacts. For instance, text contours appear more defined on Huawei, linear boundaries are better preserved on Redmi, and perceptual blurring is substantially reduced on OPPO and iPhone. These qualitative observations further underscore the importance of modeling device geometric and photometric characteristics, and highlight the superior visual fidelity enabled by our synthetic supervision pipeline.

**Overall.** Both quantitative and qualitative results consistently demonstrate the superiority of our synthetic supervision across a wide range of FI networks and smartphone platforms. By explicitly modeling cross-device geometric misalignment and photometric variation, our method enables more effective finetuning, yielding improvements in perceptual quality, structural fidelity, and generalization to unseen devices.

## A.6 Additional ablation experiment on disparity map sources

To rigorously evaluate the robustness of disparity cues in our proposed 3D-TPR encoding, we systematically ablate the disparity map sources that constitute the critical foundation for similarity computation. Employing EMA-VFI[53] backbone with RAFT-based warping[42], we conduct comparative analyses between our lightweight Lite-Mono[54] architecture and the sophisticated DepthPro[7]. The experimental results are shown in Table8.

It is crucial to emphasize that this ablation study does not directly assess the disparity variation vector itself, but rather evaluates the quality and reliability of source similarity maps utilized in constructing the 3D trajectory vectors. Experimental results on the Vimeo90K benchmark demonstrate that both

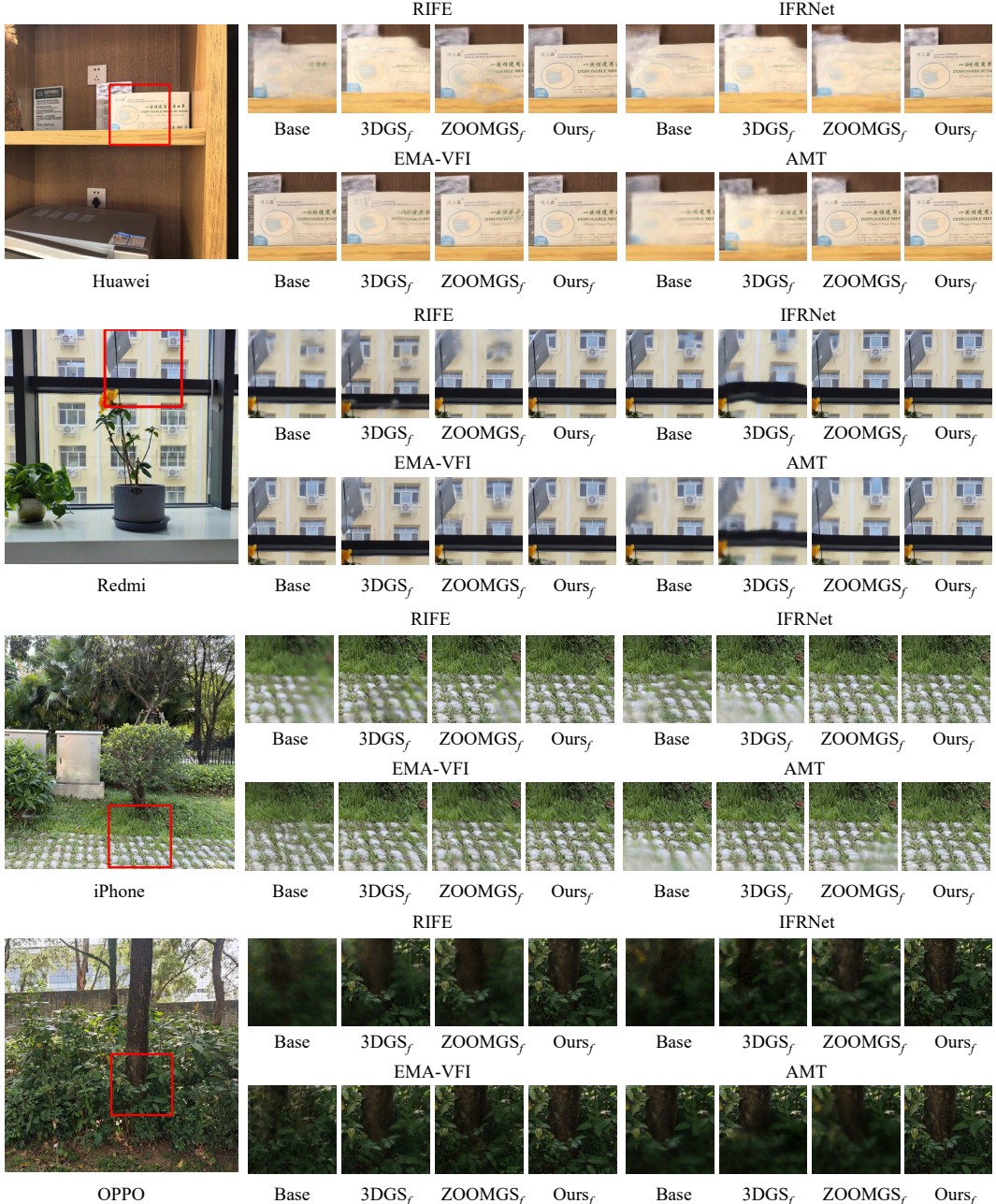

Figure 9: Qualitative comparison of finetuned interpolation results on real-world ZI tasks across four smartphone platforms and four FI networks. We compare models finetuned with virtual supervision from 3DGS, ZoomGS, and our proposed method, alongside unfinetuned baselines ("Base") implemented within the unified 3D-TPR framework. Our method consistently produces sharper structures and fewer artifacts. Notably, text contours are better preserved on Huawei, boundary lines are more clearly delineated on Redmi, and perceptual blurring is substantially reduced on iPhone and OPPO.

Table 8: Ablation study on disparity map sources.

| Method | PSNR ↑ | SSIM ↑ | LPIPS ↓ |
|---|---|---|---|
| 2D | 24.73 | 0.851 | 0.081 |
| 3D-TPR(Lite-Mono) | 24.86 | 0.853 | 0.080 |
| 3D-TPR(Depth-pro) | 24.90 | 0.852 | 0.079 |

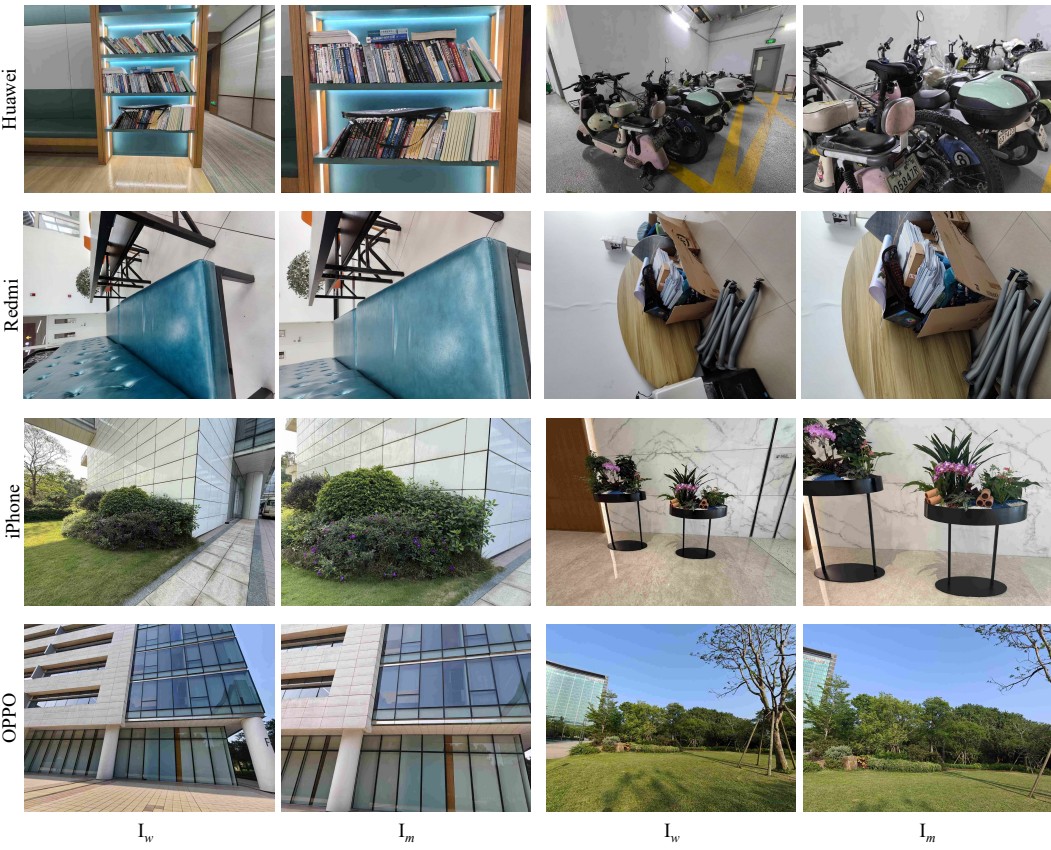

Figure 10: Representative dual-camera samples from our real-world multi-device ZI test sets. Each row corresponds to a different smartphone platform (Huawei, Redmi, iPhone and OPPO), showing paired zoom images captured under diverse scenes and lighting conditions. For layout consistency, RedmiK50Ultra samples are rotated 90 degrees counterclockwise.

3D-TPR implementations consistently surpass the 2D baseline, validating the stability and generalizability of our motion priors across different disparity estimators. Notably, while DepthPro[7] generates perceptually sharper outputs, its computational complexity yields only marginal performance gains. Consequently, we adopt Lite-Mono[54] as our preferred disparity estimator, achieving an optimal balance between training efficiency and reconstruction quality.

### A.7 Details of the real-world multi-device ZI test sets

To comprehensively evaluate the generalization ability of different ZI methods, we construct real-world multi-device test sets spanning four mainstream smartphone platforms: HuaweiPura70Ultra, RedmiK50Ultra, OPPOFindX7Ultra, and iPhone16ProMax. For each device, we collect a substantial number of dual-camera zoom sequences under diverse environmental conditions and usage scenarios.

The complete test sets consist of 499 sequences, including 160 from Huawei, 69 from OPPO, 170 from iPhone, and 100 from Redmi. These sequences span a broad range of real-world scenes, such as indoor environments (e.g., offices, shopping malls, cafés) and outdoor locations (e.g., parks, sidewalks, playgrounds), with variations in illumination, motion intensity, and zoom transitions to reflect practical deployment conditions.

Image resolution varies by device: Huawei (2580×1560), Redmi (1216×1632), iPhone (2016×1512), and OPPO (2048×1536). Each sequence captures paired wide and main camera images along the zoom trajectory, providing a challenging and diverse benchmark for evaluating ZI methods. Representative samples from the test sets are shown in Figure 10.

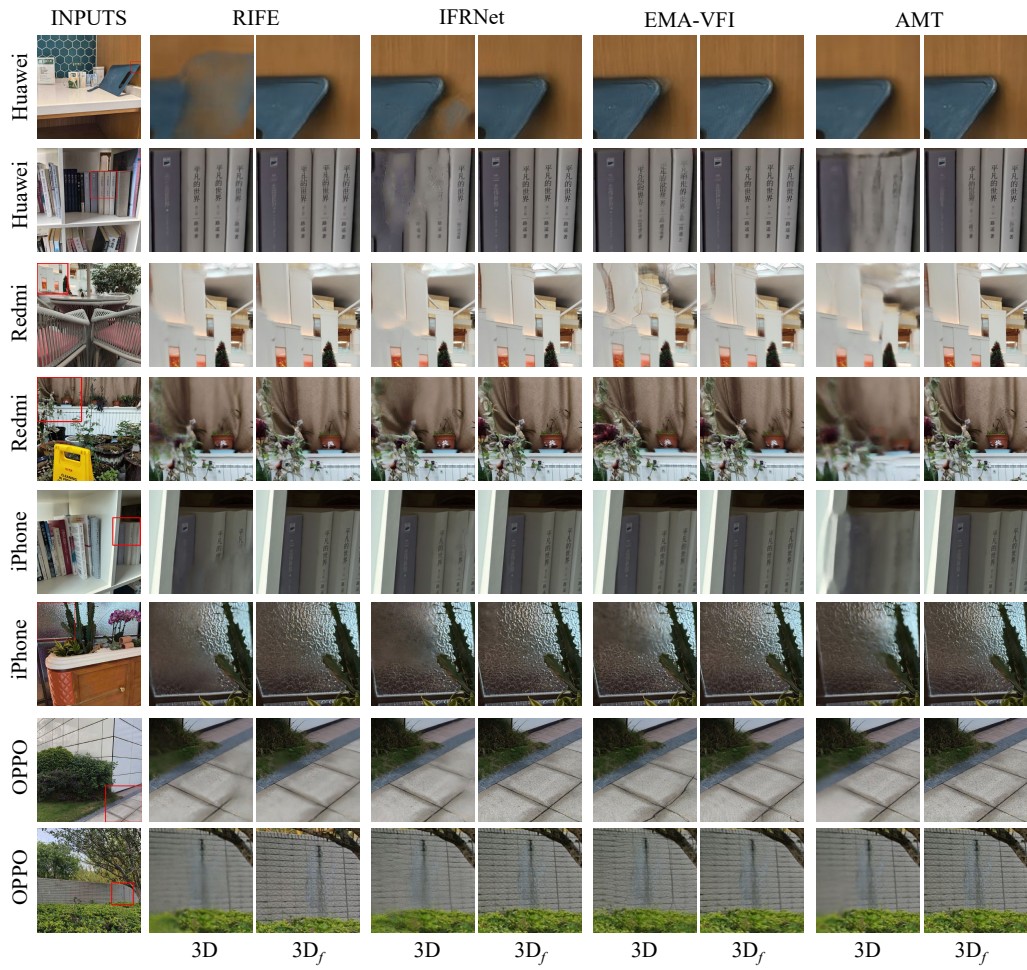

Figure 11: Additional visual comparisons on real-world ZI test data from four smartphone platforms. Subscript $_f$ indicates models finetuned on our ZI dataset. Finetuned models produce sharper structures and improved geometric consistency across diverse scenes.

## A.8   Additional visual results of OmniZoom on the real-world ZI test sets

To further evaluate the generalization capability of our method, we present additional visual comparisons on the real-world ZI test sets. Specifically, we compare 3D-TPR models before and after finetuning on the ZI dataset across four smartphone platforms. As shown in Figure 11, the finetuned model produces more temporally consistent and perceptually sharper interpolations, particularly in challenging regions involving parallax, fine textures, or low-light conditions. These results further demonstrate the effectiveness of our ZI dataset in enhancing cross-device generalization for real-world zoom interpolation.

## A.9   Benchmarking ZI dataset across 1D, 2D, and 3D-TPR frameworks

The 1D-indexed models are directly adopted from publicly released versions of each corresponding FI network. The 2D-indexed counterparts follow the interpolation strategy introduced in [60]. For each framework, we finetune the models using our proposed ZI dataset and evaluate their performance on the real-world multi-device test sets. Both qualitative and quantitative comparisons are conducted to assess the improvements enabled by ZI supervision under different indexing paradigms.

**Quantitative comparisons.** We conduct a comprehensive evaluation of the 1D, 2D, and 3D-TPR indexing frameworks across four FI models and four real-world smartphone platforms, considering both the base models and those finetuned on our ZI dataset. As shown in Table 9, the 3D-TPR framework

Table 9: Quantitative result of 1D, 2D, and 3D-TPR frameworks across four FI models and four devices. All models are evaluated before and after finetuning on our ZI dataset. **Bold** indicates the best performance after finetuning, and underline marks the best result before finetuning. 3D-TPR framework reliably achieves superior perceptual quality, and its finetuned variant ($3D_f$) outperforms others in most cases, demonstrating the effectiveness of our ZI dataset and the 3D-TPR design.

| Device | Metrics | RIFE | | | | | | IFRNet | | | | | |
|---|---|---|---|---|---|---|---|---|---|---|---|---|---|
| | | 1D | 2D | 3D | $1D_f$ | $2D_f$ | $3D_f$ | 1D | 2D | 3D | $1D_f$ | $2D_f$ | $3D_f$ |
| Huawei | NIQE↓ | 3.9181 | 3.8464 | 3.8651 | 3.8678 | 3.8170 | **3.7422** | 3.6995 | 3.6801 | 3.4852 | 3.6044 | 3.6455 | **3.5035** |
| | PI↓ | 4.2567 | 4.2505 | 4.1537 | 4.0698 | 4.0599 | **3.9360** | 4.0141 | 3.9570 | 3.3150 | 3.6448 | 3.6136 | **3.2520** |
| | CLIP-IQA↑ | 0.4078 | 0.3691 | 0.4939 | 0.4573 | 0.4858 | **0.5233** | 0.4910 | 0.5422 | 0.5784 | 0.5649 | 0.5770 | **0.5909** |
| | MUSIQ↑ | 49.4314 | 44.8268 | 58.8362 | 58.4623 | 58.7098 | **60.8585** | 51.2033 | 57.4632 | 73.0233 | 61.2380 | 63.1001 | **74.1220** |
| Redmi | NIQE↓ | 5.2253 | 5.0138 | 4.8764 | 4.6925 | 4.5118 | **4.4720** | 5.3165 | 5.0098 | 4.2837 | 4.6223 | 4.5292 | **4.0695** |
| | PI↓ | 5.6387 | 5.3615 | 5.0247 | 4.8289 | 4.6331 | **4.5195** | 5.5147 | 5.1108 | 3.5993 | 4.6454 | 4.5053 | **3.3880** |
| | CLIP-IQA↑ | 0.3661 | 0.4077 | 0.4664 | 0.4559 | 0.4851 | **0.4930** | 0.3691 | 0.4219 | 0.4913 | 0.4765 | 0.4947 | **0.5152** |
| | MUSIQ↑ | 51.7491 | 56.3870 | 61.9235 | 60.6873 | 62.0812 | **63.6990** | 51.3791 | 57.5453 | 73.7573 | 61.7805 | 62.4252 | **74.3871** |
| iPhone | NIQE↓ | 4.3433 | 4.2031 | 4.1027 | 3.9339 | 3.9532 | **3.8601** | 4.0136 | 3.8786 | 3.6923 | 3.7891 | 3.8169 | **3.6953** |
| | PI↓ | 4.6718 | 4.5340 | 4.3503 | 4.1674 | 4.2135 | **4.0657** | 4.4137 | 4.2090 | 3.3942 | 4.0014 | 3.9636 | **3.3165** |
| | CLIP-IQA↑ | 0.4476 | 0.4821 | 0.5363 | 0.5371 | 0.5345 | **0.5492** | 0.4814 | 0.5240 | 0.5829 | 0.5666 | 0.5778 | **0.5949** |
| | MUSIQ↑ | 52.2715 | 55.0577 | 59.6800 | 59.2367 | 58.8978 | **60.7988** | 52.7781 | 57.4088 | 73.3569 | 60.8860 | 61.9734 | **73.8292** |
| OPPO | NIQE↓ | 5.2103 | 4.6364 | 4.6100 | 5.0142 | 4.5575 | **4.3968** | 5.9164 | 4.5820 | 5.2039 | 5.6720 | 5.7078 | **5.1133** |
| | PI↓ | 5.4442 | 5.0848 | 5.0104 | 5.0965 | 5.0650 | **4.6815** | 5.1909 | 4.9749 | 4.9603 | 5.0287 | 4.8435 | **4.7144** |
| | CLIP-IQA↑ | 0.4270 | 0.4989 | 0.5366 | 0.5450 | 0.5699 | **0.5830** | 0.5181 | 0.5525 | 0.5212 | 0.5207 | **0.5667** | 0.5461 |
| | MUSIQ↑ | 54.8990 | 63.2645 | 65.7912 | 63.3583 | 65.0204 | **67.1494** | 54.8757 | 67.2635 | 75.2718 | 64.9371 | 67.2739 | **75.4545** |

| Device | Metrics | EMA-VFI | | | | | | AMT | | | | | |
|---|---|---|---|---|---|---|---|---|---|---|---|---|---|
| | | 1D | 2D | 3D | $1D_f$ | $2D_f$ | $3D_f$ | 1D | 2D | 3D | $1D_f$ | $2D_f$ | $3D_f$ |
| Huawei | NIQE↓ | 4.6877 | 3.6363 | 3.8470 | 3.7737 | **3.4852** | 3.8341 | 4.7231 | 3.5665 | 3.6459 | 3.9866 | 3.6246 | **3.4547** |
| | PI↓ | 3.7060 | 3.7240 | 3.8566 | **3.6638** | 3.7186 | 3.8467 | 4.9499 | 3.5147 | 3.6132 | 3.9974 | 3.4639 | **3.3121** |
| | CLIP-IQA↑ | 0.5619 | 0.5612 | 0.5621 | 0.5588 | 0.5535 | **0.5684** | 0.3127 | 0.5428 | 0.5498 | 0.5898 | 0.5944 | **0.6018** |
| | MUSIQ↑ | 60.4395 | 61.7263 | 62.7403 | 62.8008 | 63.0230 | **63.5048** | 53.6305 | 71.4369 | 71.7016 | 73.4803 | 72.7456 | **73.7092** |
| Redmi | NIQE↓ | 4.5060 | 4.4088 | 4.3424 | 4.4864 | 4.3537 | **3.8852** | 5.9112 | 5.1925 | 4.7426 | 4.9700 | 5.1082 | **3.9835** |
| | PI↓ | 4.6410 | 4.6708 | 4.5974 | 4.5829 | 4.5665 | **4.5519** | 5.0623 | 5.4335 | 4.0235 | 4.8863 | 5.0036 | **3.3548** |
| | CLIP-IQA↑ | 0.4651 | 0.4599 | 0.4807 | 0.4809 | 0.4812 | **0.4947** | 0.4191 | 0.4336 | 0.4754 | 0.5082 | 0.4944 | **0.5383** |
| | MUSIQ↑ | 55.3260 | 57.0371 | 59.6610 | 60.2753 | 59.9715 | **60.7579** | 49.2505 | 57.6671 | 71.9982 | 63.6003 | 61.2496 | **74.3325** |
| iPhone | NIQE↓ | 4.9410 | 4.8155 | 4.0555 | 4.7661 | 4.2139 | **4.0934** | 5.0436 | 3.5932 | 3.7053 | 3.5792 | 3.5883 | **3.5087** |
| | PI↓ | 5.0171 | 3.9994 | 4.1304 | 4.5345 | **3.8462** | 4.1123 | 5.0410 | 3.4293 | 3.5367 | 3.3379 | 3.3233 | **3.2661** |
| | CLIP-IQA↑ | 0.4603 | 0.5352 | 0.5387 | 0.5367 | 0.5462 | **0.5503** | 0.3723 | 0.5931 | 0.5930 | 0.6064 | 0.6120 | **0.6211** |
| | MUSIQ↑ | 55.5380 | 58.1422 | 58.6204 | 57.6496 | 59.1077 | **60.1002** | 57.8120 | 71.9805 | 72.1243 | 73.5609 | 72.9853 | **73.6150** |
| OPPO | NIQE↓ | 5.0104 | 4.5568 | 4.5166 | 4.9704 | 4.5293 | **4.5071** | 5.7811 | 4.9042 | 5.3017 | 4.7389 | 4.7885 | **4.6699** |
| | PI↓ | 5.1828 | 4.9569 | 4.9022 | 5.0729 | 4.9006 | **4.8752** | 5.8036 | 4.9676 | 5.3808 | 5.0513 | 4.9911 | **4.5702** |
| | CLIP-IQA↑ | 0.5820 | 0.5622 | 0.5590 | 0.5071 | 0.5502 | **0.5591** | 0.3499 | 0.4798 | 0.4518 | 0.4673 | 0.5461 | **0.5595** |
| | MUSIQ↑ | 67.0395 | 67.4327 | 67.5921 | 67.1736 | 67.5282 | **67.6444** | 54.9076 | 73.0842 | 72.1563 | 72.8543 | 74.7868 | **75.2187** |

consistently outperforms its 1D and 2D counterparts after finetuning, achieving the best performance (highlighted in bold) across nearly all perceptual quality metrics. Notably, even without finetuning, 3D-TPR frequently ranks second (underlined), demonstrating strong generalization capability and architectural robustness. In contrast, 1D and 2D variants exhibit only limited performance gains, particularly under challenging cross-device settings. These results support two conclusions: (1) while our ZI dataset provides performance gains across all indexing paradigms, the improvements are most pronounced under the trajectory-aware 3D-TPR framework; and (2) OmniZoom (i.e., $3D_f$-TPR) offers a universal, plug-and-play solution for real-world cross-device zoom interpolation, effectively bridging domain gaps and enhancing perceptual quality across diverse hardware platforms.

**Qualitative comparisons.** Beyond quantitative evaluation, we provide qualitative results in Figures 12 and 13, illustrating ZI outputs on Huawei, Redmi, iPhone, and OPPO devices across four FI networks. Each figure presents six variants per network: 1D, 2D, and 3D, along with their finetuned counterparts ($1D_f$, $2D_f$, and $3D_f$).

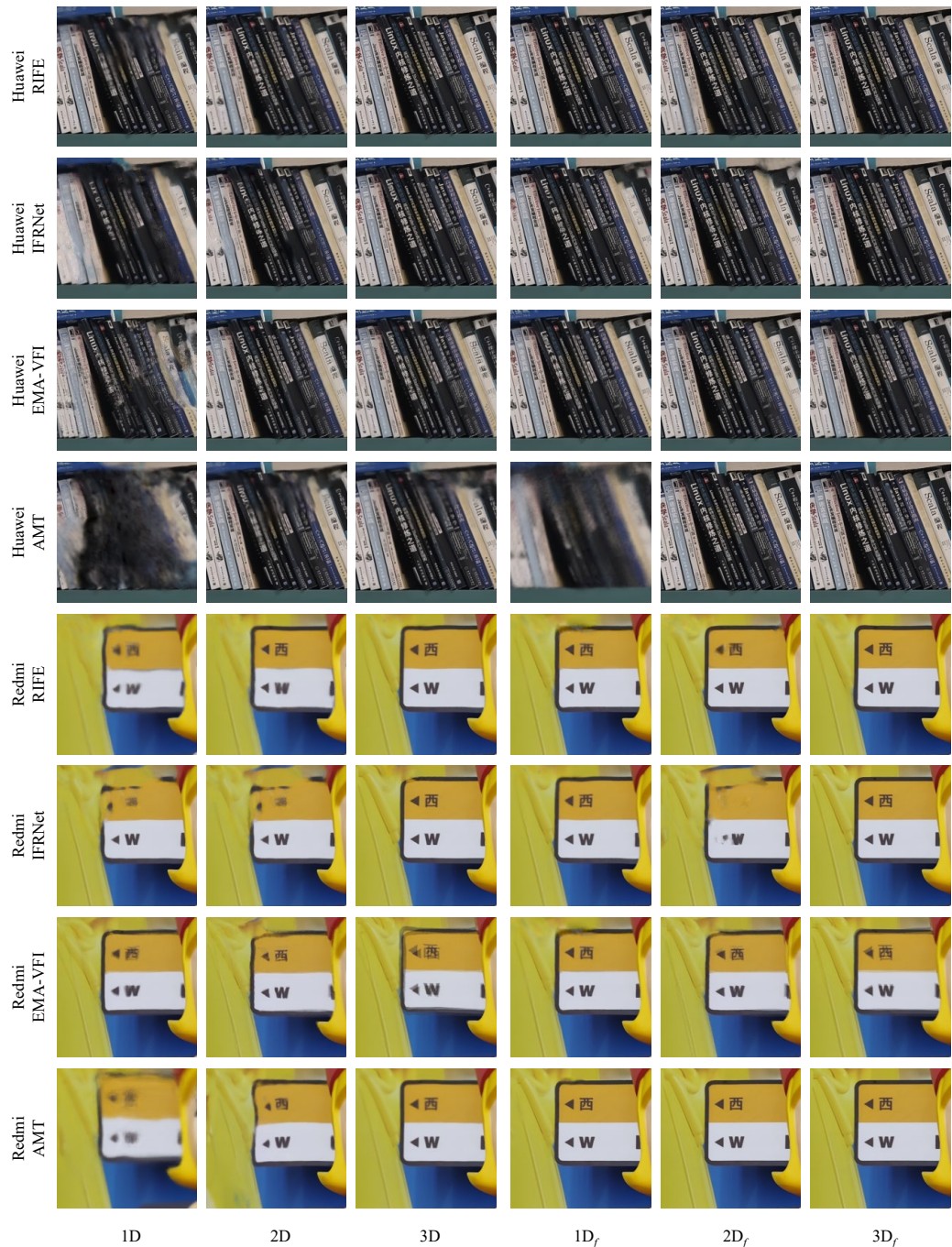

Figure 12: Qualitative comparisons on Huawei (top four rows) and Redmi (bottom four rows). While 1D and 2D models exhibit noticeable blurring and detail loss across networks, 3D$_f$ effectively restores fine details, producing sharper structures and improved local contrast.

Several key observations emerge from these visualizations. First, the 3D configuration consistently produces sharper and more coherent results than its 1D and 2D counterparts, highlighting the effectiveness of the 3D-TPR framework in real-world ZI tasks. Second, across all indexing paradigms, all finetuned models show clear perceptual improvements, demonstrating the broad applicability of our ZI dataset across diverse FI architectures.

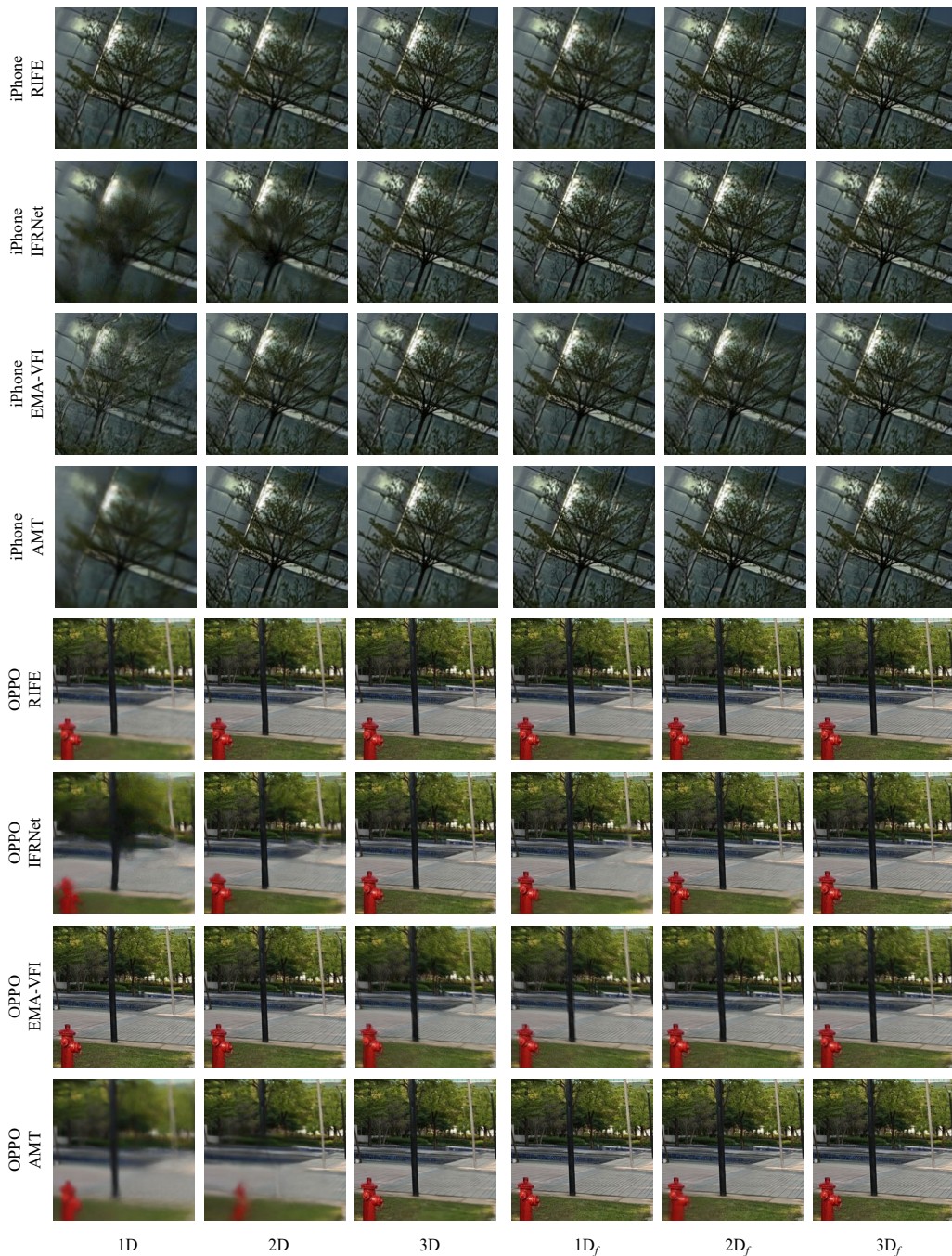

Figure 13: Qualitative comparisons on iPhone and OPPO test sets under rich structural conditions. While 1D and 2D variants exhibit edge instability and perceptual artifacts, $3D_f$ produces noticeably clearer results with sharper contours and reduced blurring, effectively preserving structural fidelity.

Notably, $3D_f$ achieves the highest perceptual quality in nearly all cases, with fewer artifacts and enhanced texture fidelity. For example, on Redmi sequences, $3D_f$ recovers sharper character contours and fine line structures compared to other variants. On iPhone and OPPO, where scenes contain more intricate textures and geometric structures, 1D and 2D variants frequently introduce ghosting and edge artifacts, issues that are substantially mitigated by the $3D_f$ configuration.

These results underscore the strength of OmniZoom as a unified, cross-device solution for plug-and-play zoom interpolation, offering robust perceptual quality across various hardware platforms.

Table 10: Upper bound evaluation of the proposed 3D-TPR framework on four FI networks using ground-truth 3D supervision. The 2D framework serves as a baseline [60]. All experiments on 3D use the 3D trajectory progress ratio as timestep map. 3D-t-m indicates using only 3D-TPR encoding; 3D-m adds texture-focus strategy; 3D-t includes mask penalty constraint; and 3D integrates all components. **Bold** indicates improvements over 2D baseline.

| Network | RIFE | | | IFRNet | | | EMA-VFI | | | AMT | | |
|---|---|---|---|---|---|---|---|---|---|---|---|---|
| | PSNR↑ | SSIM↑ | LPIPS↓ | PSNR↑ | SSIM↑ | LPIPS↓ | PSNR↑ | SSIM↑ | LPIPS↓ | PSNR↑ | SSIM↑ | LPIPS↓ |
| 2D | 27.4050 | 0.9010 | 0.0863 | 27.1276 | 0.8994 | 0.0780 | 24.7313 | 0.8514 | 0.0810 | 27.1726 | 0.9017 | 0.0807 |
| 3D-t-m | **28.7510** | **0.9267** | **0.0773** | **28.2484** | **0.9225** | **0.0715** | **25.4078** | **0.8618** | **0.0780** | **28.4850** | **0.9269** | 0.0837 |
| 3D-m | **28.7311** | **0.9264** | **0.0764** | **28.3486** | **0.9241** | **0.0690** | **25.4705** | **0.8631** | **0.0800** | **28.5942** | **0.9285** | **0.0767** |
| 3D-t | **28.5518** | **0.9234** | **0.0764** | **28.1917** | **0.9206** | **0.0686** | **25.3557** | **0.8618** | **0.0781** | **28.5895** | **0.9277** | 0.0810 |
| 3D | **28.6548** | **0.9255** | **0.0773** | **28.2967** | **0.9220** | **0.0656** | **25.4984** | **0.8633** | **0.0770** | **28.6271** | **0.9286** | **0.0768** |

## A.10 Upper bound analysis of 3D-TPR framework

To evaluate the performance ceiling of the 3D-TPR framework, we adopt ground-truth 3D geometry as the timestep map, thereby ensuring perfect consistency between training and inference. As shown in Table 10, we conduct a complementary ablation study under this setting for fair comparison.

Results indicate that utilizing consistent 3D similarity maps during both training and testing leads to substantial gains across all metrics and networks: PSNR improves by over 1.0 dB on average, SSIM consistently increases, and LPIPS is significantly reduced. These results reveal the latent capacity of the 3D-TPR framework when accurate geometry is available throughout.

This upper bound analysis provides valuable insight into the theoretical capacity of our method. While full 3D supervision may not be available in practice, our empirical findings suggest that leveraging lightweight geometry estimation modules or substituting with physically obtainable priors (e.g., depth from stereo, SLAM-based pose, or monocular depth) can approximate similar benefits. This observation offers practical guidance for future extensions that aim to balance reconstruction quality and computational feasibility.

## A.11 Additional visual results of 3D-TPR framework

We present additional qualitative comparisons between the proposed 3D-TPR framework and conventional 2D baselines, as shown in Figure 14. The results demonstrate that 3D-TPR effectively restores degraded regions in 2D-based outputs, particularly in areas suffering from motion-induced blur where structural details are severely corrupted or entirely missing.

Moreover, 3D-TPR alleviates misalignment artifacts commonly encountered in 2D models when handling textures with high levels of repetition and self-similarity, such as linear structures or dot-like patterns, by incorporating trajectory-aware spatial priors. In addition to these localized corrections, our method consistently produces globally improved visual quality, exhibiting sharper details, better structural coherence, and a more realistic overall appearance. These findings further highlight the advantages of integrating 3D spatial awareness into temporal interpolation tasks.

## A.12 Broader impacts

This work is motivated by the goal of enhancing frame interpolation in zoom scenarios, with direct benefits for a wide range of applications in mobile photography, video conferencing, augmented reality (AR), and digital content creation. Our proposed framework improves the temporal smoothness and spatial consistency of interpolated video frames across heterogeneous smartphone platforms, contributing to better visual quality in real-world capture workflows. In particular, it enables smoother zoom transitions without requiring any hardware-level changes due to zero cost, offering cost-effective enhancements for existing consumer devices.

Such capabilities are especially impactful in mobile videography, where users often experience abrupt visual discontinuities during zoom. Our method can help reduce motion artifacts and maintain semantic coherence in challenging scenes, which is beneficial for video-based communication,

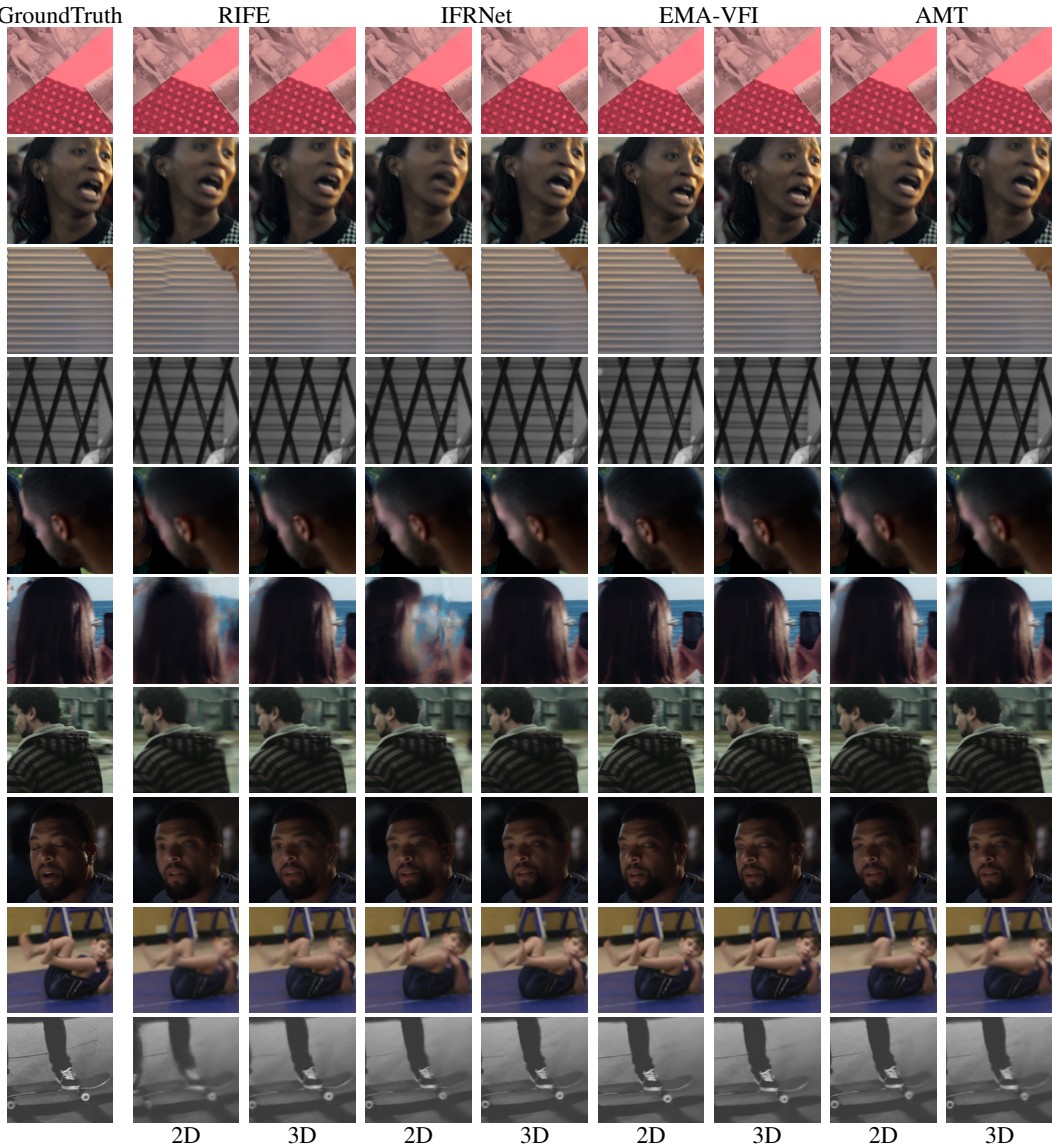

GroundTruth    RIFE    IFRNet    EMA-VFI    AMT

2D    3D    2D    3D    2D    3D    2D    3D

Figure 14: Visual comparison of 2D and 3D interpolation results across networks at the same timestep. The 3D method consistently yields less-blurred and neater outputs across all networks.

content sharing on social platforms, and immersive AR experiences that rely on high frame fidelity. Furthermore, cross-device compatibility and plug-and-play integration facilitate broader adoption across hardware vendors, making advanced interpolation accessible without retraining for each model.

Nevertheless, as with other generative video technologies, improvements in visual realism may pose risks if misused, such as enabling the creation of misleading or manipulated video content. While our framework is restricted to supervised frame interpolation under dual-camera settings and does not support arbitrary video generation, we emphasize the importance of responsible use. To mitigate misuse, we do not release general-purpose generative models, and our dataset is limited to constrained zoom interpolation scenarios. We encourage future efforts to pair such technologies with appropriate safeguards and transparency mechanisms to ensure ethical deployment.

