# OpenReview forum: "OmniZoom: A Universal Plug-and-Play Paradigm for Cross-Device Smooth Zoom Interpolation"
_NeurIPS.cc/2025/Conference — NeurIPS 2025 poster_

### Official Review · Reviewer_Viu6 · 2025-06-04

**Clarity:** 3
**Significance:** 3
**Originality:** 2
**Rating:** 4
**Confidence:** 4

**Summary:**

Target at disparities in intrinsic/extrinsic parameters and different image processing pipelines, this paper proposes a cross-device virtual data generation method. Experimental results verify the effectiveness of the proposed method.

**Questions:**

1. Why Spatial transition modelling do not rely on device-specific assumptions? More explanation or visual illustrations should be given.
2. For 3D-TPR framework, how to set the parameter t?
3. From Fig. 2, the differences among the wide-angle image, the synthesized intermediate frame and the main camera image are not clear.
4. From Fig. 3, 2D seems to achieve better performance than 3D-TPR for some small parts, or says that I cannot easily find the difference between them. The authors should give more explanations.
5. How about the complexity and running speed of the proposed method? Can the proposed method be applied in reality directly?
6. 3D-TPR uses RAFT to estimate 2D pixel-wise optical flows. How much impact will the quality of optical flow have on the experimental results? Is the time cost of optical flow estimation acceptable?

**Ethical Concerns:**

["NO or VERY MINOR ethics concerns only"]

**Final Justification:**

My concerns have been addressed in the rebuttal period, so I decided to raise my final rating to Borderline accept. I want to point out that the orgnaisation of the paper, esepicially the tables and figures should be refined in the final version.

**Limitations:**

Yes. I think this method has no negative societal impact .

**Quality:**

2

**Strengths And Weaknesses:**

Strengths
1. The paper is generally well-organised and easy to follow.
2. The targeted problem is valuable, a plug-and-play paradigm for zoom interpolation is of practical value.
3. This paper effectively articulates the issues associated with interpolation, providing a clear and comprehensive explanation from my point.
4. A dataset is proposed by the authors, which can benefits the research community, and the experimental parts are thorough.

Weaknesses
1. The proposed modules in the paper are somewhat fragmented, and the innovations in each module should be strengthened.
2. It is difficult to directly see the advantages of the method from the resulting images. The author could consider using more representative graphs or zooming in on local details.
3. Some details of visual results can be enlarged to make them clearer, such as Fig. 3 and Fig. 4.
4. A limitation part can be included in the paper to make this research more comprehensive.

---

> ### Author Rebuttal · Authors · 2025-07-30
>
> - **W1: On the fragmentation and innovation of proposed modules**
>
> Thank you for the insightful comment. While our framework involves multiple components, they are **tightly coupled** to address a core challenge in cross-device zoom interpolation: **(1)** the lack of high-quality, device-agnostic supervision, and **(2)** the inherent motion ambiguity in standard FI pipelines.
>
> **To address (1)**, we design a device-agnostic data generation pipeline with three coordinated modules. We enhance this process with **spatial transition modeling** and **dynamic color adaptation** to decouple geometric and photometric differences between devices. Furthermore, a **cross-domain consistency learning scheme** ensures the image maintains semantic alignment and photometric fidelity across devices. Together, these modules form a unified pipeline for generating high-quality, device-agnostic supervision. The key advantages of our data generation strategy are summarized in the table below:
> |**Method**|**Intermediate frame synthesis**|**Color transition (device-agnostic)**|**Spatial transition (device-agnostic)**|**Semantic consistency**|**Cross-device generalization**|
> |---|:---:|:---:|:---:|:---:|:---:|
> |3DGS|✓|✗|✗|✗|✗|
> |ZoomGS|✓|✗|✗|✗|✗|
> |**Ours**|✓|✓|✓|✓|✓|
>
> **To address (2)**, we propose 3D-TPR to enhance zoom interpolation's motion reasoning and perceptual quality, introducing a **3D trajectory encoder** for monocular-disparity-aware correspondence, a **texture-focus** loss emphasizing high-frequency details, and **mask uncertainty penalty** suppressing ambiguous predictions in occluded/low-confidence regions, collectively improving robustness and perceptual fidelity. The innovations of our strategies are shown in the following table.
>
> |**Method**|**1D temporal indexing**|**2D spatial indexing**|**3D depth indexing**|**Preserves texture details**|**Suppresses occlusion ambiguities**|
> |---|:---:|:---:|:---:|:---:|:---:|
> |Conventional FI|✓|✗|✗|✗|✗|
> |InterpAny|✓|✓|✗|✗|✗|
> |**3D-TPR (Ours)**|✓|✓|✓|✓|✓|
> - **W2 & W3: Clarification on visual advantages and figure improvements.**
>
> We thank the reviewer for this valuable suggestion. We would like to clarify that the visual improvements, particularly ghosting reduction and structure preservation, are already evident in the current results. For example, in Fig. 4, our method produces noticeably clearer outputs in the Huawei power outlet, OPPO grass, and fine lines in the Redmi and iPhone samples. These perceptual gains are also supported by improved no-reference metrics in Table 4.
>
> We agree that enlarging key regions would better highlight these improvements. Due to rebuttal constraints, we cannot update the figures here, but in the camera-ready version, we will revise Figures 3 and 4 by adding zoomed-in patches and more representative examples for clarity.
>
> - **W4: Regarding the presentation of limitations.**
>
> Thank you for the suggestion. Due to space constraints, we included our limitations in Appendix A.10. We agree that making them more visible improves transparency, and will move the key points into the main paper in the camera-ready version.
>
> Our main limitations are: (1) minor adjustments to training settings (e.g., batch size or loss weights) may be needed for different FI backbones; and (2) while our dataset spans multiple smartphones, it could be further expanded to handle more extreme device variations. That said, our method still generalizes well, as shown by its strong performance on unseen platforms.
>
> - **Q1: Regarding the device-agnostic nature of spatial transition modeling.**
>
> We thank the reviewer for raising this important point. Our spatial transition module is explicitly designed to be device-agnostic. It models zoom-related camera motion solely based on the **relative transformation between the wide and main cameras**, without relying on any actual device-specific assumptions such as known baselines, factory calibration data, or distortion priors. We estimate camera intrinsics and extrinsics via COLMAP and interpolate between the two estimated poses to construct a continuous trajectory. As shown in Table 2 and Figure 4, this design leads to strong cross-device generalization.
>
> - **Q2: Clarifying *t* in 3D-TPR framework.**
>
> We sincerely thank the reviewer for this thoughtful question.
>
> **During training**, `t = Sim(i, j, k), ∀ (i, j, k)`. `Sim(i, j, k)` is derived from ground-truth frames using RAFT and Lite-Mono (see Equation 13).
>
> **During inference**, the ground-truth frame is not accessible, the exact similarity map cannot be computed. Following the strategy used in prior work such as InterpAny, it is sufficient to provide a *t = fixed-timestep map* (mentioned in line 237 in main text) in the same manner as conventional time-indexing methods.
> However, the semantics of this indexing map have shifted from an uncertain timestep map to a more **deterministic motion hint**. Physically, this encourages the model to move each object at constant speeds along their trajectories. In practice, this constant-speed assumption serves as **a valid approximation for smooth zoom scenarios commonly found in mobile photography.**
>
> As shown in Section 4, although this may lead to pixel-level misalignments relative to ground-truth, it still yields significantly improved perceptual quality. This suggests that the model, once trained with motion-aware 3D-TPR supervision, can robustly generalize to uniform motion priors during test time. Indeed, this training-inference strategy is not unique to FI task, but for a wide range of machine learning problems. In some areas, researchers have come up with similar methods [1][2]. We will clarify this design choice in the revised version.
>
> We will clarify this design choice in the revised version.
>
> - **Q3: Regarding the clarity of Figure 2.**
>
> Thank you for the observation. We agree that the transitions may not be immediately noticeable due to resolution and layout constraints in Figure 2. To clarify, one clear example is the first triplet from the Redmi row. For example, in the first Redmi triplet, the green plant in the background shows a gradual color shift, and the scarf’s position changes consistently across views, indicating a plausible transition path.
>
> In the camera-ready version, we will enlarge key regions and add annotations in Figure 2 to better highlight these transitions.
>
> - **Q4: Clarifying visual differences between 2D and 3D-TPR in Fig. 3.**
>
> We appreciate this opportunity to clarify: while overall frames may appear similar, 3D-TPR consistently improves challenging regions (fine structures / motion boundaries / occlusions) where ghosting or blurring artifacts occur.
>
> **Fig. 3 examples:**
>
> - **RIFE (row 1):** 3D-TPR better preserves tree stripe patterns vs 2D blur.
> - **IFRNet (row 2):** More natural fingers in 3D-TPR vs 2D smearing.
> - **AMT-S (row 3):** Guitar strings remain sharp and aligned in 3D-TPR vs 2D misalignment.
> - **EMA-VFI (row 4):** Clearer bracelet reflections in 3D-TPR vs 2D detail loss.
>
> We also provide additional examples in Appendix Fig. 14. In the revised version, we will clearly highlight these areas to better guide the reader’s attention. These localized enhancements reflect our 3D-aware design's strength, even when full-frame metrics may not show large differences.
>
> - **Q5: Complexity and Real-world Applicability**
>
> Thank you for this important point. Our method is **plug-and-play**: all components apply only during training. As shown in Q2, we revert to a uniform map at inference, introducing **no extra inference cost** and **preserving the base model's original inference speed/memory footprint**, ensuring full mobile/cloud compatibility.
>
> - **Q6: On the impact of optical flow quality on the experimental results.**
>
> We thank the reviewer for this valuable question. To analyze optical flow quality's impact on our 3D-TPR framework, we conduct a controlled ablation by downsampling RAFT inputs to different resolutions (1×/2×/4×/8×). As RAFT is resolution-sensitive, this setup offers an effective proxy for flow fidelity variation.  All estimated flows are bilinearly upsampled to original resolution for training.
>
> On Vimeo90K with AMT and IFRNet, lower-resolution flows (4×/8×) lead to noticeable degradation (e.g., up to 1.2dB PSNR drop), while 2× performs nearly on par with full-resolution 1× but reduces memory and compute significantly. We thus adopt 2× as a practical efficiency and accuracy trade-off.
>
> Importantly, RAFT is used only during training, with an average cost of 27.5ms per image pair at 2×. No flow estimation is needed during inference. These results confirm 3D-TPR’s robustness: it maintains strong performance even under moderately degraded flow input.
>
>
> |Benchmarks|Metrics|IFRNet 1x|IFRNet 2x|IFRNet 4x|IFRNet 8x|AMT 1x|AMT 2x|AMT 4x|AMT 8x|
> |---|---|---|---|---|---|---|---|---|---|
> |Vimeo90k|PSNR ↑|27.21|27.21|26.01|25.01|27.25|27.22|26.01|24.94|
> ||SSIM ↑|0.901|0.901|0.867|0.847|0.902|0.902|0.849|0.832|
> ||LPIPS ↓|0.070|0.074|0.085|0.094|0.083|0.084|0.106|0.127|
>
> **Reference**
>
> [1] Y. Wang, D. Stanton, Y. Zhang, R.-S. Ryan, E. Battenberg, J. Shor, Y. Xiao, Y. Jia, F. Ren, and R. A. Saurous, “Style tokens: Unsupervised style modeling, control and transfer in end-to-end speech synthesis,” in International conference on machine learning. PMLR, 2018, pp.5180–5189.
>
> [2] Y. Xu, H. Tan, F. Luan, S. Bi, P. Wang, J. Li, Z. Shi, K. Sunkavalli, G. Wetzstein, Z. Xu et al., “Dmv3d: Denoising multi-view diffusion using 3d large reconstruction model,” arXiv preprint arXiv:2311.09217, 2023.

---

### Official Review · Reviewer_qPjC · 2025-06-24

**Clarity:** 3
**Significance:** 3
**Originality:** 3
**Rating:** 5
**Confidence:** 4

**Summary:**

The paper presents a method by which view interpolation can be tailored to better address the multi-camera view interpolation problem for small baseline capture devices.  In particular, this work focuses on extending existing feedforward models by proposing a replacement positional encoding scheme based on flow and depth information.

This work also generalizes the data generation and data modeling approaches from ZoomGS to be applicable to more than just previously calibrated devices.  In particular, a color calibration network better models the variation in color calibration between lenses for better training data generation.

The authors claim the following contributions:
* Novel pipeline for ZI data generation
* 3D-TPR - encodings and losses that can be inserted into sota ZI model frameworks
* OmniZoom - the synthesis of these components into a common framework

**Questions:**

I need clarification on how the similarity map encoding is computed with 3D-TPR at time t if I_t is not known until after generation at test time.

**Ethical Concerns:**

["NO or VERY MINOR ethics concerns only"]

**Final Justification:**

My concerns have been addressed.  I will keep my rating of accept.

**Limitations:**

The limitations are relegated to the appendices and are not substantially explored.

**Paper Formatting Concerns:**

formatting is fine

**Quality:**

3

**Strengths And Weaknesses:**

+ The paper is quite well written.  The flow definitely relies on knowledge of ZoomGS early on and some exposition on how it relates would have been helpful.  In particular, I felt Figure 1 has difficulty standing on its own.

+ Ablations are presented, albeit relegated to the appendix.

= I think all the numbers don't do the task justice.  Clearly the qualitative results show reduced ghosting, but the numbers have very small differences.  We're maybe getting to the point where some additional statistical analysis is required to prove these results are significant.

- The disparity variation vector's definition seemed arbitrary and alternatives were not compared in the paper.

- I'm having a difficult time understanding what I_t is at test time in order to give it as input to 3D-TPR.  This is necessary to compute V_{0->t} and Disp_t.  Perhaps V_{0->1} is scaled and Disp_0 and Disp_1 are interpolated using the flow field?

- There's no ablation comparing ColorNet to simpler photometric calibration methods.

---

> ### Author Rebuttal · Authors · 2025-07-30
>
> - **W1: Regarding the clarity of relation to ZoomGS and Figure 1.**
>
> Thank you for the helpful comment. While our data generation is inspired by ZoomGS’s use of 3DGS for intermediate frame synthesis, this influence is limited to **the use of 3DGS as a data generation tool**. Our framework is independently designed and introduces three novel components:
> (1) spatial transition modeling,  (2) color adaptation, and  (3) cross-domain consistency learning,  which are absent in ZoomGS.
> Figure 1 illustrates our proposed pipeline. Its current version may not be fully self-explanatory. Due to NeurIPS rebuttal constraints, we will revise this figure in the camera-ready version with clearer module labels and updated captions.
>
> - **W2: On the significance of quantitative gains and the need for statistical analysis.**
>
> We thank the reviewer for the insightful suggestions. Due to page limits, ablation results were initially in the appendix. We appreciate your attention and will move expanded results to the main paper for better visibility and clarity in the camera-ready version.
>
> Although improvements on pixel-centric metrics such as PSNR and SSIM appear numerically modest, the qualitative results (e.g., reduced ghosting in Fig. 3-4) demonstrate **clear perceptual benefits**.  We argue that in most real-world applications, the goal of VFI  is not to produce pixel-aligned ground truth reconstructions, but to **synthesize plausible frames with high perceptual quality**. Furthermore, PSNR and SSIM are less sensitive to common VFI artifacts such as blur and ghosting [1], which our design explicitly targets by **resolving velocity ambiguity**. So, pixel-centric metrics are not always informative in evaluating perceptual quality.
> To better assess perceptual quality, we evaluate three perceptual metrics: PI, MUSIQ, and FLOLPIPS [2], on Vimeo90K using EMA-VFI. These human-aligned metrics highlight each component’s strengths. Notably, **FLOLPIPS is specifically designed for video frame interpolation,** and it uniquely incorporates **motion consistency priors** via flow-guided weighting.
>
> As shown below, our method consistently outperforms 2D baseline across all metrics.
> **Importantly, our design is not a naive stacking of modules.** Each component is intentionally designed to target specific and well-known limitations in Zoom Interpolation: **motion ambiguity**, **texture degradation**, and **occlusion ambiguities**. Their effects are distinct, measurable, and complementary:
>
> - The **3D-TPR encoding** (3D-t-m) enhances motion consistency over the 2D baseline, reflected by a clear improvement in **FLOLPIPS** (0.128 → 0.119). This reduction suggests that incorporating 3D trajectory cues helps mitigate motion ambiguity and improves flow-aligned perceptual coherence.
>
> - The **texture-focus strategy** (3D-m) achieves the most significant drop in **FLOLPIPS** (0.128 → 0.113) and the largest gain in **MUSIQ** (55.968 → 57.362) among individual modules, demonstrating its strong ability to preserve high-frequency details, particularly those poorly captured by 2D baselines.
>
> - The **mask uncertainty penalty** (3D-t) contributes the most to reducing **PI** among single components (5.071 → 4.934), indicating its effectiveness in suppressing occlusion artifacts and improving perceptual sharpness.
>
> When integrating all three modules into the complete 3D-TPR design (**3D full**), the results in the best overall performance across all metrics. This confirms that each module contributes uniquely, and their combination yields a **more perceptually robust and accurate interpolation pipeline**.
>
> |**Method**|**PSNR ↑**|**SSIM ↑**|**LPIPS ↓**|**PI ↓**|**MUSIQ ↑**|**FLOLPIPS ↓**|
> |---|---|---|---|---|---|---|
> |2D|24.73|0.851|0.081|5.071|55.968|0.128|
> |3D-t-m|24.78|0.852|0.081|5.041|56.615|0.119|
> |3D-m|24.74|0.852|0.084|5.022|57.362|0.113|
> |3D-t|24.80|0.853|0.081|4.934|56.946|0.116|
> |3D (full)|**24.86**|**0.853**|**0.080**|**4.802**|**58.725**|**0.091**|
>
> - **W3: Regarding the rationale and alternatives of the disparity variation vector.**
>
> We thank the reviewer for the thoughtful question. First, we clarify that the disparity variation vector $Var^{z} = \Delta D_{0\rightarrow t} / \Delta D_{0\rightarrow 1}$ is **not used during training**, nor does it directly affect training or inference. Its role is purely conceptual: to illustrate the notion of normalized motion progression in the depth dimension, complementing the 2D motion ratio $Var^{xy} = V\^{xy-plane}\_{0\rightarrow t} / V\^{xy-plane}\_{0\rightarrow 1}$  in the xy-plane. Our actual 3D-TPR computation does not explicitly use this vector. Instead, we compute cosine similarity between 3D displacement vectors constructed from estimated depth and flow.
>
> To evaluate the robustness of disparity cues in our 3D-TPR encoding, we **ablated disparity map sources** – the critical factor for similarity computation. Using fixed EMAVFI backbone and RAFT warping, we compared our lightweight Lite-Mono against advanced DepthPro [3].
>
> |**Method**|**PSNR ↑**|**SSIM ↑**|**LPIPS ↓**|
> |---|---|---|---|
> |2D|24.73|0.851|0.081|
> |3D-TPR (Lite-Mono)|24.86|0.853|0.080|
> |3D-TPR (Depth-pro)|24.90|0.852|0.079|
>
> *Note that this ablation does not evaluate the disparity variation vector per se, but instead assesses the source disparity maps used in constructing the 3D trajectory vectors.* Vimeo90K results show both 3D-TPR versions consistently outperform the 2D baseline, confirming stable, generalizable motion priors regardless of disparity estimator. Despite sharper DepthPro outputs, its complexity yields only marginal gains. We adopt Lite-Mono for optimal training efficiency-quality trade-off.
>
> - **W4 & Q: Clarification on similarity map input at inference.**
>
> We sincerely thank the reviewer for this thoughtful question.
> **During training**, the similarity map (replacing fixed-timestep map) is derived from ground-truth frames using RAFT and Lite-Mono (see Equation 13).
>
> **During inference**, $I_t$ is not accessible, and the exact similarity map cannot be computed. Following the strategy used in prior work such as InterpAny [4], it is sufficient to provide a *uniform map* (mentioned in line 237 in the main text). In the same manner as conventional time-indexing methods, a fixed-timestep map at time $t$ can be used.
> However, the semantics of this indexing map have shifted from an uncertain timestep map to a more **deterministic motion hint**. Physically, this encourages the model to move each object at constant speeds along their trajectories. In practice, this constant-speed assumption serves as **a valid approximation for smooth zoom scenarios commonly found in mobile photography.**
>
> As shown in Section 4, although this may lead to pixel-level misalignments relative to ground-truth, it still yields significantly improved perceptual quality. This suggests that the model, once trained with motion-aware 3D-TPR supervision, can robustly generalize to uniform motion priors during test time.  Indeed, this training-inference strategy is not unique to FI task, but for a wide range of machine learning problems. In some areas, researchers have come up with similar methods [5][6].
>
> We will clarify this design choice in the revised version. Moreover, we have observed that if exact similarity map is used as input for inference(Appendix A.8), frame interpolation can yield gains. Therefore, we also aim to employ more lightweight methods in our subsequent work to obtain a coarse map to enhance performance.
>
> - **W5: Ablation on ColorNet.**
>
> Thank you for the helpful suggestion. We conducted a no-reference image quality comparison between our ColorNet and common photometric calibration baselines:
> (1) No Calibration,  (2) Mean-Std Normalization [7], and  (3) Reinhard Color Transfer [8].
> Evaluations were performed on rendered intermediate frames using NIQE, PI, CLIP-IQA, and MUSIQ.
>
> As shown below, **ColorNet consistently outperforms all baselines**, highlighting its advantage in modeling fine-grained photometric variations that simpler methods cannot capture. We will include these ablations in the revised manuscript to support our design choice.
>
> |**Method**|**NIQE ↓**|**PI ↓**|**CLIP-IQA ↑**|**MUSIQ ↑**|
> |---|---|---|---|---|
> |No Calibration|3.5832|3.8378|0.5193|68.4036|
> |Mean-Std Norm|3.3193|3.7757|0.4577|63.0220|
> |Reinhard Transfer|3.2357|3.7345|0.4588|64.3218|
> |**ColorNet (Ours)**|**2.8885**|**3.1515**|**0.6099**|**69.3328**|
> - **Limitations: Clarification on limitations' placement and content:**
>
> We thank the reviewer for pointing this out. Due to the page limit of the main paper, we included the limitations in Appendix A. We will integrate them into the main paper in the camera-ready version.
>
> Our main limitations are:
> (1) while OmniZoom supports plug-and-play usage across different FI backbones, mild adjustments to training schedules may improve results in practice; and  (2) our dataset can be further enriched to improve robustness under extreme device variations.
>
> Reference
>
> [1] The unreasonable effectiveness of deep features as a perceptual metric. In Proceedings of the IEEE conference on computer vision and pattern recognition
>
> [2] Flolpips: A bespoke video quality metric for frame interpolation.
>
> [3] Depth pro: Sharp monocular metric depth in less than a second. In International Conference on Learning Representations.
>
> [4] Clearer frames, anytime: Resolving velocity ambiguity in video frame interpolation. In European Conference on Computer Vision.
>
> [5] Style tokens: Unsupervised style modeling, control and transfer in end-to-end speech synthesis, in International conference on machine learning.
>
> [6] Dmv3d: Denoising multi-view diffusion using 3d large reconstruction model.
>
> [7] Perceptual losses for real-time style transfer and super-resolution. In European Conference on Computer Vision.
>
> [8] Color transfer between images. IEEE Computer Graphics and Applications.

---

### Official Review · Reviewer_9EWN · 2025-06-30

**Clarity:** 2
**Significance:** 2
**Originality:** 2
**Rating:** 3
**Confidence:** 3

**Summary:**

The paper proposes a deep learning framework for smooth transitions across differently configured cameras, such as dual cameras in smartphones. While prior work on frame interpolation partially addresses smooth transitions between frames, key challenges remain: (i) zoom interpolation lacks suitable ground-truth intermediate frames, as collecting such data requires continuous zoom using dual-camera hardware; and (ii) motion ambiguity, which has been addressed by prior work such as InterpAny. To overcome data scarcity, the proposed framework leverages 3D Gaussian Splatting to synthesize virtual intermediate frames. The second issue (motion ambiguity) is addressed by a 3D Trajectory Progress Ratio (3D-TPR) mechanism that uses a pixel-wise similarity map to reduce artifacts. Experimental results show some improvement on real-world data.

**Questions:**

Thank you for submitting your work to NeurIPS 2025.
The challenge addressed in this paper is important, particularly for enabling seamless experiences on multi-camera mobile devices, which are increasingly ubiquitous.
However, there are three major concerns related to the novelty of the proposed framework, the result, and the clarity of the writing.

1. The core technique —using a pixel-wise similarity map-- appears closely related to the approach introduced in InterpAny. To highlight the novelty of this work, it would be helpful to clearly articulate how the proposed method differs conceptually or technically from InterpAny. In particular, I suggest authors to clearly explain how the "motion ambiguity" dealt in this paper is different from that dealt in InterpAny, and how the core technique resolves this challenge.
2. A key limitation is that the proposed framework integrates multiple components--3D-TPR encoding, texture-focus strategy, and mask penalty constraint--yet the overall performance gain is modest and not the best among ablations (Appendix Table 5), especially compared to the improvement achieved by InterpAny. Furthermore, the results do not include error bars or statistical significance measures, making it difficult to assess the robustness of the improvements. Clarifying the rationale for including all components and providing evidence of statistical significance over baselines would strengthen the empirical claims.
3. The paper uses several under-explained terms, which makes it difficult to fully understand the proposed approach:
  - "Velocity ambiguity" is mentioned without definition or citation. Although it appears once in the InterpAny paper, the lack of a direct reference and explanation makes it unclear what the challenge entails or how 3D-TPR addresses it.
  - "Fixed-timestep map" is another term not found in the referenced literature and should be briefly clarified.
  - The paper also does not explain the justification for how decoupling geometric and photometric characteristics remove device-dependent parameters; an explaination would help clarify this important design choice.

**Ethical Concerns:**

["NO or VERY MINOR ethics concerns only"]

**Final Justification:**

I raised my final rating as 3 from 2. Revision version adressed my concerns about the method's minimal performance gains by providing additional evaluations on perceptual quality.

**Limitations:**

yes

**Paper Formatting Concerns:**

No Issue

**Quality:**

3

**Strengths And Weaknesses:**

- Strengths:
  - The paper introduces a novel and interesting problem named Zoom Interpolation.
  - The paper includes thorough experiments, including ablation studies, to demonstrate the framework's performance under diverse conditions.
  - The proposed method effectively integrates with and improves four different frame interpolation modules.

- Weaknesses:
  - The "motion ambiguity" that the authors aim to address in Zoom Interpolation is not clearly explained, making it difficult to fully understand the specific challenge and how the proposed techniques effectively resolve it. As a result, the perceived novelty of the techniques is diminished.
  - The integrated components offer limited performance gains without statistical evidence, and their inclusion lacks clear justification.
  - Several terms (e.g., velocity ambiguity, fixed-timestep map, ...) are insufficiently explained, making the method hard to interpret.

---

> ### Author Rebuttal · Authors · 2025-07-30
>
> - **W1 & Q1: Clarification of "Motion Ambiguity" and Novelty Compared to InterpAny.**
>
> We thank the reviewer for this important question.  While both our method and InterpAny [1] adopt pixel-wise similarity maps, our core idea is to reformulate motion progress estimation **in the full 3D space** by leveraging monocular depth, offering a more **physically grounded** and **geometrically discriminative** alternative to tackle velocity ambiguity. We clarify the terminology and highlight the key differences below.
>
> **1. Clarifying Terminology:**
>
> **Fixed-timestep map:**  The fixed-timestep map is typically used as **the temporal input for interpolation models**, where the start and end frames are assigned time values of 0 and 1, respectively. It represents the specific time value for the frame to be inserted. For example, interpolating the middle frame corresponds to a map filled with 0.5. This 1D time prior, however, lacks object-specific motion awareness.
>
> **Velocity ambiguity:**  As defined in InterpAny, velocity ambiguity refers to **the inherent uncertainty in object motion trajectories between two frames**, due to the absence of intermediate observations. A given pixel may follow multiple plausible paths (e.g., curved vs. linear motion) or exhibit non-uniform speeds (e.g., due to foreshortening). Since fixed-timestep maps provide uniform temporal priors, interpolation models cannot resolve such ambiguity per object.
>
> **2. Comparison with InterpAny**
>
> **InterpAny's approach:**  InterpAny tackles velocity ambiguity by replacing the fixed-timestep map with a 2D distance index map, computed from the similarity between optical flows. This encourages the model to infer spatially adaptive motion progress during training.
> However, this formulation is limited to motion in the xy-plane. As shown in Table 2 and Figure 3 in the main text, relying solely on 2D optical flow leads to **indistinguishable distance indices for geometrically distinct 3D motions**, thereby **failing to resolve velocity ambiguity in realistic scenes**.
>
> *For instance, in zoom scenarios where objects exhibit foreshortened motion (i.e., moving toward or away from the camera), InterpAny’s 2D similarity map fails to differentiate fronto-parallel versus depth-variant trajectories, leading to ghosting or temporal drift. **Our 3D-TPR explicitly addresses this case.***
>
> **Our 3D-TPR approach:**  We extend the 2D distance index into 3D space by incorporating depth variation. Specifically, we estimate monocular disparity map and compute the 3D trajectory vectors between the target and end frames. Their cosine similarity yields a **3D trajectory progress ratio (3D-TPR)**, which better **reflects true physical motion and resolves velocity ambiguity in both spatial and depth dimensions.** This 3D similarity map replaces the fixed-timestep map, just as in InterpAny, but provides richer geometric priors. The formulation is fully differentiable and **can be plugged into existing VFI models without architectural changes**, applicable to existing VFI models as a plug-in.
>
> **3. Additional Contributions Beyond InterpAny**
>
> - We propose a **gradient-coupled texture-focus strategy** that enhances sharpness in high-frequency regions by reweighting the loss using image gradients. This design allows texture regions to receive stronger supervision without relying on pre-trained perceptual networks (Appendix A.2.2).
>
> - We introduce a **mask uncertainty penalty** to regularize ambiguous fusion masks in bidirectional interpolation. This leads to improved sharpness near motion boundaries and promotes faster and more stable convergence during training (Appendix A.2.3).
>
> **4. Summary of Differences**
>
> The table below compares InterpAny and our 3D-TPR framework across major aspects. Our method introduces three main improvements:
> (1) We extend motion modeling into **3D space** using 3D-TPR encoding, which better distinguishes depth-related motions.
> (2) A **texture-focus strategy** that enhances detail preservation in high-frequency regions.
> (3) A **mask uncertainty penalty** to incentivize perceptually sharper frame synthesis.
>
> |**Method**|**1D temporal indexing**|**2D spatial indexing**|**3D depth indexing**|**Preserves texture details**|**Suppresses occlusion ambiguities**|
> |---|:---:|:---:|:---:|:---:|:---:|
> |InterpAny|✓|✓|✗|✗|✗|
> |**3D-TPR**|✓|✓|✓|✓|✓|
>
> - **W2 & Q2: On the necessity of all components and the significance of overall performance gain.**
>
> We sincerely thank the reviewer for these valuable comments. Although improvements on pixel-centric metrics such as PSNR and SSIM appear numerically modest, the qualitative results (e.g., reduced ghosting and enhanced sharpness in Fig. 3 and Fig. 4) demonstrate **clear perceptual benefits**.
> We argue that in most real-world applications, the goal of video frame interpolation (VFI) is not to produce pixel-aligned ground truth reconstructions, but to **synthesize plausible intermediate frames with high perceptual quality**. Furthermore, PSNR and SSIM are less sensitive to common VFI artifacts such as blur and ghosting [2], which our design explicitly targets by **resolving velocity ambiguity**. As such, pixel-centric metrics are not always informative in evaluating perceptual quality.
>
> To better reflect perceptual quality, we evaluate three additional perceptual metrics: **PI**, **MUSIQ**, and **FLOLPIPS** [3] on the Vimeo90K benchmark using EMA-VFI. Due to space constraints, we include results from one representative network here, while full results will be provided in the revised paper. These metrics are more aligned with human judgment and highlight the strengths of each component.
> Notably, **FLOLPIPS is specifically designed for video frame interpolation,** and it uniquely incorporates **motion consistency priors** via flow-guided weighting. This makes it particularly appropriate for evaluating perceptual quality and temporal coherence in smooth zoom interpolation.
>
> As shown below, our method consistently outperforms InterpAny (2D baseline) across all metrics.
> **Importantly, our design is not a naive stacking of modules.** Each component is intentionally designed to target specific and well-known limitations in Zoom Interpolation: **motion ambiguity**, **texture degradation**, and **occlusion ambiguities**. Their effects are distinct, measurable, and complementary:
>
> - The **3D-TPR encoding** (3D-t-m) enhances motion consistency over the 2D baseline, reflected by a clear improvement in **FLOLPIPS** (0.128 → 0.119). This reduction suggests that incorporating 3D trajectory cues helps mitigate motion ambiguity and improves flow-aligned perceptual coherence.
>
> - The **texture-focus strategy** (3D-m) achieves the most significant drop in **FLOLPIPS** (0.128 → 0.113) and the largest gain in **MUSIQ** (55.968 → 57.362) among individual modules, demonstrating its strong ability to preserve high-frequency details, particularly those poorly captured by 2D baselines.
>
> - The **mask uncertainty penalty** (3D-t) contributes the most to reducing **PI** among single components (5.071 → 4.934), indicating its effectiveness in suppressing occlusion artifacts and improving perceptual sharpness.
>
> When integrating all three modules into the complete 3D-TPR design (**3D full**), the results in the best overall performance across all metrics. This confirms that each module contributes uniquely, and their combination yields a **more perceptually robust and accurate interpolation pipeline**. These improvements are **statistically significant ($p < 0.05$)**, as verified by paired $t$-tests over all test samples.
>
> |**Method**|**PSNR ↑**|**SSIM ↑**|**LPIPS ↓**|**PI ↓**|**MUSIQ ↑**|**FLOLPIPS ↓**|
> |---|---|---|---|---|---|---|
> |2D|24.73|0.851|0.081|5.071|55.968|0.128|
> |3D-t-m|24.78|0.852|0.081|5.041|56.615|0.119|
> |3D-m|24.74|0.852|0.084|5.022|57.362|0.113|
> |3D-t|24.80|0.853|0.081|4.934|56.946|0.116|
> |3D (full)|**24.86**|**0.853**|**0.080**|**4.802**|**58.725**|**0.091**|
>
> - **W3 & Q3: Explanation of Key Terms:**
>
> Velocity Ambiguity and Fixed-timestep Map explanation see in W1 & Q1.
>
> **Justification for Geometric–Photometric Decoupling**
>
> Our decoupling of geometric and photometric characteristics is a central component for enabling cross-device generalization in zoom interpolation. Below, we clarify how this design removes device-dependent parameters:
>
> **Geometric decoupling**
> Our spatial transition module operates entirely on *relative camera transformations*, using estimated intrinsics and extrinsics from the input wide and main images. It does not assume any actual device-specific parameters such as fixed baselines, known calibration, or distortion models. Thus, geometry is modeled purely through camera pose interpolation, independent of the device configuration. This ensures generalizability across devices with varying hardware setups.
>
> **Photometric decoupling**
> Instead of relying on handcrafted calibration curves or device-specific tone mappings, our ColorNet module learns *content-adaptive color corrections* from the input pair. Each image is encoded into a latent embedding, and ColorNet predicts per-Gaussian color offsets to align appearance across views. This data-driven design allows the system to adaptively handle color differences without assuming any pre-defined device behavior.
>
> **Summary**
> By disentangling geometric transitions from hardware configuration, and photometric alignment from device-specific color priors, our framework avoids hardcoded assumptions and achieves **device-agnostic generalization** (see Table 2 and Figure 4). We will further clarify this rationale in the revised manuscript.
>
> [1] Clearer frames, anytime: Resolving velocity ambiguity in video frame interpolation.
>
> [2] The unreasonable effectiveness of deep features as a perceptual metric.
>
> [3] Flolpips: A bespoke video quality metric for frame interpolation.

---

> > ### Comment · Reviewer_9EWN · 2025-08-06
> >
> > Thank you for your detailed response. My concerns have been well addressed, and I appreciate your efforts in conducting additional evaluations to demonstrate the method's perceptual effectiveness. I will raise my final rating.

---

> > > ### Author Response · Authors · 2025-08-07
> > >
> > > We're happy to hear that we've managed to alleviate your concerns. Thanks again for the review, and if there are any outstanding concerns that we can address, please let us know.

---

### Official Review · Reviewer_Wduj · 2025-07-03

**Clarity:** 2
**Significance:** 2
**Originality:** 2
**Rating:** 4
**Confidence:** 3

**Summary:**

OmniZoom presents a universal plug-and-play paradigm for cross-device smooth zoom interpolation. This paper defines zoom interpolation as a critical task for dual-camera smartphones to synthesize geometrically and photometrically consistent intermediate frames during zoom transitions. The primary challenges addressed are: 1) the significant geometric and photometric inconsistencies between cameras, 2) the lack of suitable ground-truth intermediate frames for training, and severe motion ambiguity caused by complex, and 3) non-linear object motion and parallax effects inherent in zoom transitions. OmniZoom tackles these issues through two key design insights: first, a novel cross-device virtual data generation method utilizing 3D Gaussian Splatting. Second, it introduces a plug-and-play 3D Trajectory Progress Ratio (3D-TPR) framework that encodes 3D motion-aware correspondence, integrating a texture-focus strategy for high-frequency detail preservation and mask penalty constraints to suppress interpolation artifacts. The evaluation highlights leading performance on various smartphone platforms.

**Questions:**

What is the supported input resolution of OmniZoom? Based on the details of FI models, it seems the FI models are pre-trained on a resolution of 448x256. Is this the target output resolution of OmniZoom? If so, please explain why this relatively low resolution is suitable for mobile photography. If not, please specify the target resolution and explain if there is any mismatch between the pretraining resolution and fine-tuning resolution. If the mismatch exists, I would also like to know if that could lead to any model performance issues.

**Ethical Concerns:**

["NO or VERY MINOR ethics concerns only"]

**Final Justification:**

The author has sufficiently discussed the limitations and missing technical details. Given the potential broad impact of the introduced work, I'm still leaning accepting. My final rating score remain unchanged.

**Limitations:**

One key component of the proposed work is the generated synthetic data. Based on the paper, there were 205 sequences generated, and
16 frames each. Considering the paper aims to support cross-device smooth zoom support, it is hard to be convinced that this data scale can truly support the goal. A deeper discussion about the data generation details would be great. I am also curious if any potential data augmentation steps could be integrated.

**Quality:**

3

**Strengths And Weaknesses:**

- The work shows promising testing results on real-world images from various mobile phones.
- The framework is built on top of established dual-camera smooth zoom designs and provides extended support in visual qualities.
- The paper is well structured and written.

---

> ### Author Rebuttal · Authors · 2025-07-30
>
> - **Questions:**
>
> Thank you very much for this insightful question. We sincerely appreciate your attention to detail and your concern about the resolution settings, which are indeed crucial for evaluating the practicality and scalability of our framework in real-world mobile photography scenarios.
>
> We would like to clarify that during pretraining, we used the Vimeo-90K septuplet dataset with a native resolution of 448×256. During finetuning, our synthetic Zoom Interpolation dataset contains higher-resolution samples: 1632×1224 for Redmi and 2133×1600 for Huawei. However, we adopt **a patch-based training strategy** in both stages, consistent with the original design of each FI model: we use 256×256 patches for EMAVFI, and 224×224 for RIFE, AMT, and IFRNet, identical to their official training protocols. Therefore, there is **no resolution mismatch** between the pretraining and finetuning stages.
>
> *At inference time*, **OmniZoom supports arbitrary-resolution inputs and can seamlessly adapt to the native resolutions of real mobile devices**, including 1632×1224 (Redmi), 2133×1600 (Huawei), 2016×1512 (iPhone), and 2048×1536 (OPPO). Although these resolutions are higher than the training patches, we observe **no noticeable degradation in performance or visual quality** for two key reasons:
>
>   - **All FI models used are fully convolutional**, and therefore generalize naturally to higher resolution inputs without any structural modification or performance degradation.
>
>   - **The Zoom Interpolation task exhibits continuous and smooth camera motion**, with predominantly local spatial variations. As such, models trained on moderate-sized patches are capable of capturing high-resolution local correspondences effectively at test time.
>
> We greatly appreciate this question as it allowed us to better explain this design detail, and we will revise the paper accordingly to ensure clarity for readers.
>
> ---
>
> - **Limitations:**
>
> We sincerely thank the reviewer for raising this important point regarding data scale and its role in enabling cross-device smooth zoom interpolation.
>
> While our dataset consists of 205 synthetic zoom sequences (16 frames each) captured from two devices, we emphasize that its design **prioritizes structural diversity and cross-device generalizability over raw volume**. Specifically, we focus on capturing **geometric and photometric variations** that commonly arise in real-world zoom operations across devices.
>
> Our device-agnostic supervision pipeline includes:
>
>   - **Device-agnostic spatial transition modeling:** We interpolate both intrinsic and extrinsic parameters between wide and main camera views to simulate smooth and geometrically coherent virtual viewpoints, independent of device-specific assumptions.
>
>   - **Photometric decoupling via color adaptation:** A learnable ColorNet predicts per-Gaussian color offsets to adapt to photometric shifts between devices, ensuring robust rendering across camera styles.
>
>   - **Cross-domain consistency constraints:** We enforce semantic and textural consistency between rendered and real images, enhancing realism and semantic alignment.
>
>   - **Extensive data augmentation:** We apply random resizing, cropping, horizontal/vertical flipping, rotation, temporal reversal, channel permutation, and pose jittering to enrich the diversity of spatial and photometric variations. These augmentations empirically improve the model’s robustness under diverse conditions.
>
> In addition, all models are initialized from strong pretrained FI backbones (e.g., RIFE, EMAVFI) and **finetuned with our synthetic supervision**. This significantly reduces the demand for large-scale data while ensuring effective adaptation to zoom-specific motion.
>
> As shown in Table 2 and Figure 4, our models **generalize well to unseen devices** (e.g., iPhone, OPPO), validating the transferability of our supervision design. In future work, we plan to expand the dataset to include a wider variety of devices and optical settings.

---

> > ### Comment · Reviewer_Wduj · 2025-08-05
> >
> > Thank you for the response. My concers have been addressed. My final rating remain the same.

---

> > > ### Author Response · Authors · 2025-08-06
> > >
> > > We're happy to hear that we've managed to alleviate your concerns. Thanks again for the review, and if there are any outstanding concerns that we can address, please let us know.

---

### Note · Authors · 2025-08-14

Dear ACs and Reviewers,

We sincerely thank all reviewers for their constructive feedback and valuable suggestions, which have helped improve our work.

We note that **Reviewer 9EWN** has confirmed all concerns were **''well addressed''** and will **''raise final rating''**, but **this positive update does not yet reflected** in the score,  just a gentle note for your consideration before the deadline.

We are encouraged that the reviewers found the following points:

- The paper tackles a **valuable and novel**  Zoom Interpolation task with **practical importance** for dual-camera smartphones.
- Our proposed universal plug-and-play framework shows **strong cross-device generalization** and **promising results on real-world images**.
- Our 3DGS-based data generation **effectively models geometric and photometric consistency**.
- Our 3D-TPR framework improves motion reasoning, detail preservation, and artifact suppression **across multiple FI models**.
- The paper is **well-written and well-structured** with thorough experiments, ablations, and **a dataset valuable to the community**.

In response to reviewers’ feedback, we conducted additional experiments and clarifications:

**Reviewer Wduj**: Clarified arbitrary‑resolution support, strong generalization via device‑agnostic modeling, cross‑domain consistency, and diverse augmentations despite 205 sequences.

**Reviewer 9EWN**: Explained 3D‑TPR’s 3D motion modeling beyond InterpAny, texture‑focus and mask‑penalty designs, and device‑agnostic significant gains via geometric–photometric decoupling.

**Reviewer qPjC**: Clarified independence from ZoomGS, novel modules, training–inference strategy. Confirmed perceptual gains over 2D baseline, validation across disparities and ColorNet superiority.

**Reviewer Viu6**: Emphasized tight integration for device‑agnostic supervision and motion ambiguity, validated generalization of spatial transition modeling, zero inference cost, and robustness to varied optical flow.

We believe **all concerns have been fully addressed** through additional experiments and explanations. **All reviewers acknowledged that their concerns have been resolved**. We thank all reviewers for their efforts, and would greatly appreciate timely confirmation of the rating updates before the deadline.

Sincerely,

Authors.

---

### Decision · Program_Chairs · 2025-09-17

**Decision:**

Accept (poster)

**Comment:**

The paper received mostly positive but divergent reviews (1 borderline reject, 2 borderline accepts, 1 accept).

The main concern from the borderline reject primarily centered around the paper's relation to InterpAny as well as the presentation of the paper. In the rebuttal the authors directly addressed this concern noting that InterpAny's reliance on 2D optical flow can lead to difficulties with 3D scene motion and that their 3D trajectories directly attempt to overcome these problems. They additionally discuss the gradient-coupled texture-focus strategy and a mask uncertainty penalty as novelties wrt InterpAny. The other reviewers were mostly positive, noting the promising results as well as overall organization. Though it seems like there is certainly room for improvement with respect to things like the ablation studies as well as understanding how each module interacts with each other.

After reading the review, rebuttal, and discussion, I advocate for acceptance but encourage the authors to address the presentation issues identified by Reviewer Viu6.